# Interspike intervals within retinal spike bursts combinatorially encode multiple stimulus features

**Toshiyuki Ishii**[1,2,3], **Toshihiko Hosoya**[1¤]*

**1** RIKEN Center for Brain Science and RIKEN Brain Science Institute, Wako-shi, Saitama, Japan, **2** Toho University, Funabashi-shi, Chiba, Japan, **3** Department of Physiology, Nippon Medical School, Bunkyo-ku, Tokyo, Japan

¤ Current address: Biomedical Research Department, Ricoh Company Ltd, Kawasaki-shi, Kanagawa, Japan
* toshihiko.hosoya@jp.ricoh.com

**Data Availability Statement:** All data files are available at: https://osf.io/29ect/.

**Funding:** This work was supported by research funds from RIKEN to T.H. and Grant-in-Aid for Scientific Research from the Ministry of Education,

## Abstract

Neurons in various regions of the brain generate spike bursts. While the number of spikes within a burst has been shown to carry information, information coding by interspike intervals (ISIs) is less well understood. In particular, a burst with $k$ spikes has $k−1$ intraburst ISIs, and these $k−1$ ISIs could theoretically encode $k−1$ independent values. In this study, we demonstrate that such combinatorial coding occurs for retinal bursts. By recording ganglion cell spikes from isolated salamander retinae, we found that intraburst ISIs encode oscillatory light sequences that are much faster than the light intensity modulation encoded by the number of spikes. When a burst has three spikes, the two intraburst ISIs combinatorially encode the amplitude and phase of the oscillatory sequence. Analysis of trial-to-trial variability suggested that intraburst ISIs are regulated by two independent mechanisms responding to orthogonal oscillatory components, one of which is common to bursts with a different number of spikes. Therefore, the retina encodes multiple stimulus features by exploiting all degrees of freedom of burst spike patterns, i.e., the spike number and multiple intraburst ISIs.

## Author summary

Neurons in various regions of the brain generate spike bursts. Bursts are typically composed of a few spikes generated within dozens of milliseconds, and individual bursts are separated by much longer periods of silence (~hundreds of milliseconds). Recent evidence indicates that the number of spikes in a burst, the interspike intervals (ISIs), and the overall duration of a burst, as well as the timing of burst onset, encode information. However, it remains unknown whether multiple ISIs within a single burst encode multiple input features. Here we demonstrate that such combinatorial ISI coding occurs for spike bursts in the retina. We recorded ganglion cell spikes from isolated salamander retinae stimulated with computer-generated movies. Visual response analyses indicated that multiple ISIs within a single burst combinatorially encode the phase and amplitude of oscillatory light

Culture, Sports, Science and Technology (MEXT) of Japan to T.H. (19300116). The funders had no role in study design, data collection and analysis, decision to publish, or preparation of the manuscript.

**Competing interests:** The authors have declared that no competing interests exist.

sequences, which are different from the stimulus feature encoded by the spike number. The result demonstrates that the retina encodes multiple stimulus features by exploiting all degrees of freedom of burst spike patterns, i.e., the spike number and multiple intraburst ISIs. Because synaptic transmission in the visual system is highly sensitive to ISIs, the combinatorial ISI coding must have a major impact on visual information processing.

## Introduction

Understanding the rules by which neuronal spike patterns encode information is essential for investigating the complex functioning of the nervous system [1, 2]. Neurons in various brain areas generate spike bursts, which are characterized by clusters of high-frequency spikes separated by longer periods of silence [3–5]. Burst spikes typically occur within the temporal window of postsynaptic integration (dozens of milliseconds), and thereby induce synaptic responses with a higher probability than isolated single spikes [6–8]. In this regard, bursts are believed to represent an important neuronal code [7, 9, 10]. Previous analyses of burst information coding have suggested that the number of spikes within a burst [4, 11–19], the onset timing of a burst [5, 6, 20–22], and the duration of a burst [15, 23] all carry information.

Because a burst has multiple spikes, it has one or more intraburst interspike intervals (ISIs). In theory, these intraburst ISIs can carry information if, for example, they are modulated by sensory inputs. Such burst ISI coding should have significant effects on information transfer, because the efficiency of synaptic transmission is sensitive to ISIs [24]. Consistent with this idea, recent studies suggest that ISIs within bursts carry information [15, 19, 23, 25]. Although these studies have shown that the first ISI and average ISI within a burst carry information, whether multiple ISIs within a single burst carry information in a combinatorial manner has been unclear. Theoretically, bursts with $k$ spikes have $k-1$ intraburst ISIs, and these $k-1$ ISIs could encode $k-1$ independent values that represent information. Whether burst ISIs encode information in such a combinatorial manner is unknown.

In the vertebrate retina, retinal ganglion cells (i.e., the output neurons) generate spike bursts [3, 4, 26]. While the number of spikes within bursts encodes the amplitude of light intensity modulation [4], it is unknown whether intraburst ISIs encode information. In this study, using isolated salamander retinae, we investigated whether intraburst ISIs encode information regarding visual input. Our results indicate that intraburst ISIs encode oscillatory light intensity sequences that are different from the stimulus feature encoded by the spike number. When bursts contained three spikes, the two ISIs combinatorially encoded the amplitude and phase of the oscillatory components. Further analysis of the trial-to-trial variability suggested that intraburst ISIs are determined by two independent neuronal mechanisms that respond to two orthogonal oscillatory components. Collectively, our findings demonstrate that multiple ISIs within a retinal burst combinatorially encode multiple independent stimulus features that are different from that encoded by the spike number.

## Results

### Burst spike numbers encode the amplitude of light intensity modulation

We stimulated isolated larval salamander retinae using a spatially uniform visual stimulus with intensity modulation set at 30 Hz. Ganglion cell action potentials were recorded using a multi-electrode array. OFF ganglion cells, constituting the majority of the larval salamander retinal ganglion cells, generated spike bursts (Fig 1). The majority of the spikes [82.0% ± 8.7%,

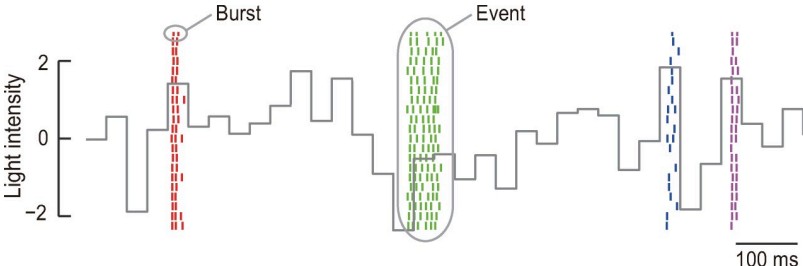

**Fig 1. Retinal ganglion cells generate reproducible bursts.** Raster plot of a single salamander OFF ganglion cell. Short vertical lines represent single spikes and each row shows spikes that occurred during a single repeat of the stimulation. Spikes of different events are shown in different colors. The gray continuous line shows the normalized light intensity of the stimulus (the mean and SD are 0 and 1, respectively).

mean ± SD (standard deviation), $n$ = 41 cells] were observed in bursts comprised of two or more spikes, indicating that multi-spike bursts represent the major retinal code.

During repeated presentation of the same stimulus, individual ganglion cells generated spike bursts at similar time-points across repeats (Fig 1) [3, 4]. This reproducibility enabled the identification of corresponding bursts across repeats, which we termed "events" (Figs 1 and S1, see Materials and Methods) [3, 4]. Bursts generated in the same event had similar numbers of spikes in different repeats of the stimulus, while those in different events often had different numbers of spikes (Fig 1). Accordingly, the number of spikes within bursts carried information about the stimulus (1.5 ± 0.3 bits per burst, mean ± SD, $n$ = 41). We next calculated the burst-triggered averages (BTAs), which represent the average stimulus sequence preceding isolated spikes and bursts with two, three, and four spikes (1-, 2-, 3-, and 4-BTA, respectively). The BTAs were sequences of different amplitudes (Fig 2A). The difference between 1-BTA and 3-BTA (3-BTA−1-BTA) had ON and OFF peaks around −170 and −40 ms relative to the burst onset (Fig 2B). The peak frequency of the 3-BTA−1-BTA was 4.7 ± 1.8 Hz (mean ± SD, $n$ = 41).

To further confirm that the number of spikes systematically encode stimulus sequences with different amplitudes (Fig 2A), we analyzed the encoding of the stimulus component represented by the sequence 3-BTA−1-BTA (Fig 2B). For this purpose we projected the stimulus sequences onto 3-BTA−1-BTA [27]. The projected values were normalized so that they have a mean = 0 and a SD = 1 ($s_{3-1}$; the dashed line in Fig 2C shows the probability density distribution $P(s_{3-1})$). The $s_{3-1}$ value at the time points of isolated spikes and bursts with 2–4 spikes had a probability distribution that was different from $P(s_{3-1})$ $P(s_{3-1}|$1-spike), $P(s_{3-1}|$2-spike), $P(s_{3-1}|$3-spike), and $P(s_{3-1}|$4-spike); solid lines in Fig 2C). The mean of these distributions was larger for a larger number of spikes (0.03 ± 0.29, 0.75 ± 0.32, 1.35 ± 0.35, and 1.66 ± 0.36, for isolated spikes, 2-spike bursts, 3-spike bursts, and 4-spike bursts, respectively, mean ± SD, $n$ = 41; Fig 2C).

Fig 2C compares the projected values sampled at all stimulus time points with those sampled at isolated spikes, 2-spike bursts, 3-spike bursts, and 4-spike bursts. To conduct comparison with the projected values sampled at all isolated spikes and bursts, we collected the projection values at the time points of the first spikes of all bursts and isolated spikes and normalized the collected values ($s_{3-1,\text{burst}}$). The dashed line in Fig 2D shows the probability density distribution $P(s_{3-1,\text{burst}})$, i.e., the distribution of $s_{3-1,\text{burst}}$ sampled all isolated spikes and bursts. $P(s_{3-1,\text{burst}}|$1-spike), $P(s_{3-1,\text{burst}}|$2-spike), $P(s_{3-1,\text{burst}}|$3-spike), and $P(s_{3-1,\text{burst}}|$4-spike) were different from $P(s_{3-1,\text{burst}})$, and the mean was larger for a larger number of spikes (−0.69 ± 0.25, 0.00 ± 0.22, 0.57 ± 0.27, and 0.87 ± 0.29, respectively, mean ± SD, $n$ = 41; Fig 2D). These results

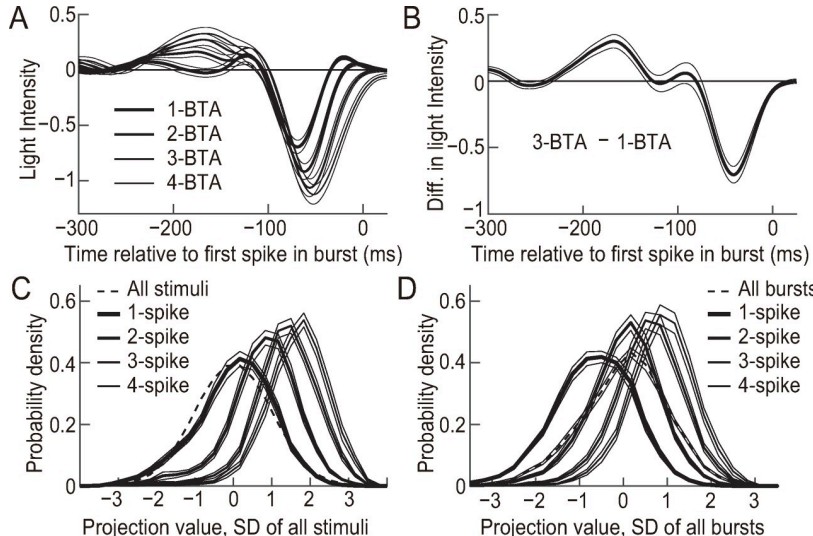

**Fig 2. The spike number within a burst encodes the amplitude of light intensity modulation. (A)** 1-, 2-, 3-, and 4-BTA indicate the average of stimulus sequences preceding isolated spikes and 2-, 3-, and 4-spike bursts, respectively. The average and standard error of the mean (SEM) were calculated across $n = 41$ cells. **(B)** Difference between 3-BTA and 1-BTA, averaged across $n = 41$ cells. The thin lines indicate the SEM calculated across the cells. **(C)** Analysis of $s_{3-1}$, the stimulus projected onto 3-BTA−1-BTA. The dashed line indicates $P(s_{3-1})$ averaged across $n = 41$ cells. The SEM calculated across for $P(s_{3-1})$ cells is not shown because it was very small. The thick lines correspond to $P(s_{3-1}|1\text{-spike})$, $P(s_{3-1}|2\text{-spike})$, $P(s_{3-1}|3\text{-spike})$, and $P(s_{3-1}|4\text{-spike})$ averaged across $n = 41$ cells. The thin lines are the SEM calculated across the cells. **(D)** Similar to (C), calculated for $s_{3-1,\text{burst}}$, which was normalized for values at all time points of isolated spikes and bursts.

indicate that the number of spikes within a burst encodes the amplitude of ON-to-OFF light intensity modulation within an interval of ~130 ms and the peak frequency of ~4.7 Hz.

## Burst ISIs encode oscillatory light intensity sequences

To investigate whether intraburst ISIs carry information, we first analyzed bursts composed of two spikes (2-spike bursts). For each ganglion cell, 2-spike bursts in the same event tended to have similar ISIs, whereas those in different events typically had different ISIs (Fig 3A and 3B). The mutual information about the stimulus encoded by 2-spike burst ISIs was 2.7 ± 0.7 bits per burst (mean ± SD, $n = 41$). These results suggest that intraburst ISIs convey information about the stimulus.

Two-spike burst ISIs exhibited little or no correlation with the average number of spikes in events (Fig 3C and 3D; correlation coefficient = 0.0 ± 0.1; mean ± SD, $n = 41$), suggesting that these ISIs were modulated according to stimulus features that were different from those that modulated the spike number. To characterize the stimulus features that modulated 2-spike burst ISIs, we extracted the stimulus sequences preceding 2-spike bursts with different ISIs (Fig 3E and 3F). The results showed that the stimulus sequences had systematic differences depending on the ISIs. For each cell, we determined the deviation from the 2-BTA by subtracting the 2-BTA (black in Fig 3F) from the average of sequences preceding the 2-spike bursts with long ISIs. The deviation from the 2-BTA was an oscillating sequence with two ON and two OFF peaks (yellow in Fig 3G). The deviation from the 2-BTA for short ISIs was the same sequence, but with the opposite sign (blue in Fig 3G). The result was similar for the majority of the neurons, with a peak frequency of 8.2 ± 3.1 Hz (mean ± SD, $n = 19$ cells that generated at least 1500 2-spike bursts; Fig 3H and S2A Fig).

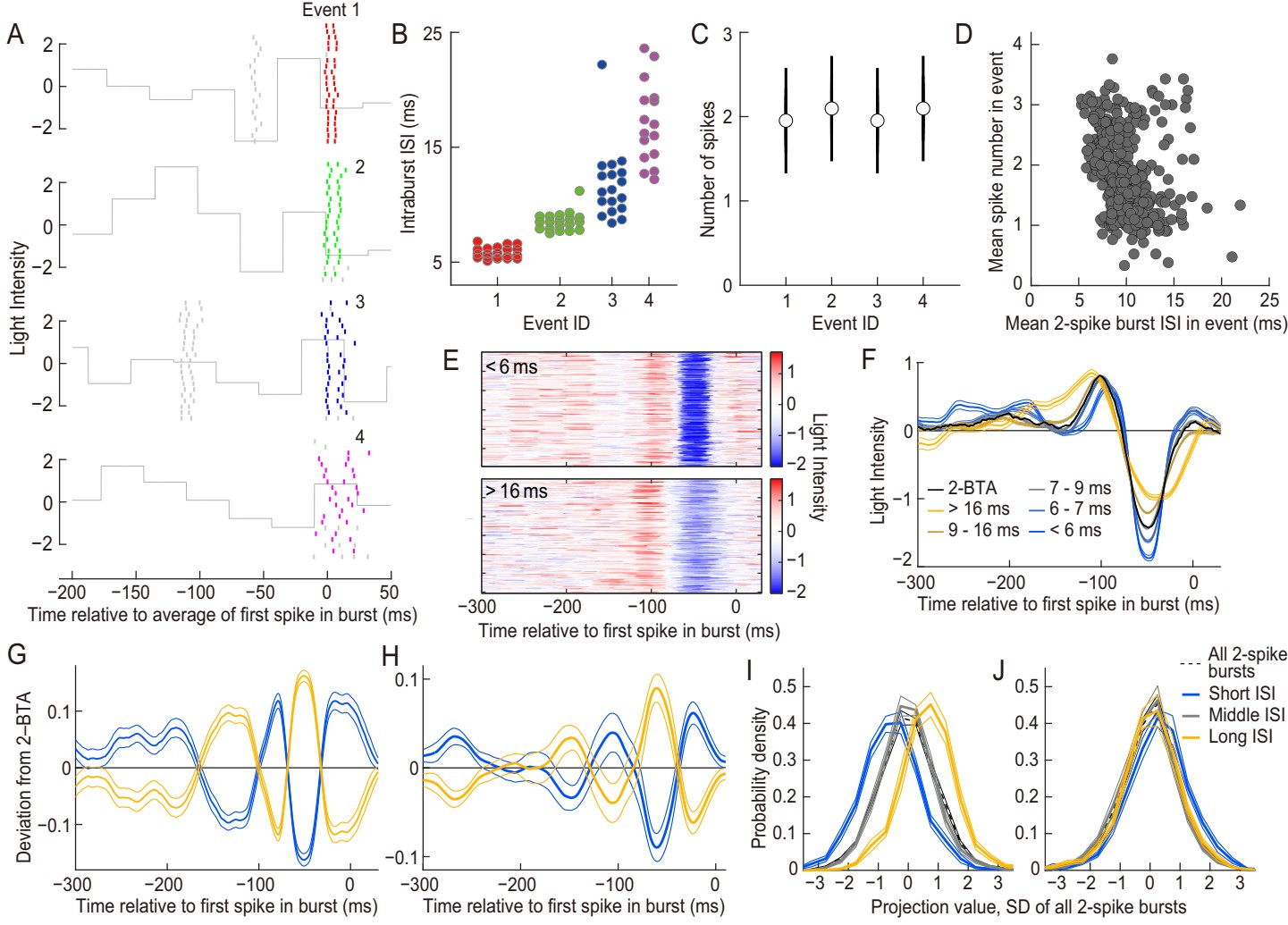

**Fig 3. Intraburst ISIs of 2-spike bursts encode an oscillatory component of the visual input. (A–G)** Data from the cell shown in Fig 1. **(A)** Raster plot. The colored lines represent 2-spike bursts. The short gray lines are the remaining spikes. The gray continuous line shows the normalized light intensity of the stimulus (the mean and SD are 0 and 1, respectively). Event IDs are shown. **(B and C)** Intraburst ISIs of 2-spike bursts **(B)** and the average and SD of the spike number **(C)** of the four events shown in (A). **(D)** Relationship between 2-spike burst ISIs and the mean spike number. Each dot represents an event with at least one 2-spike burst. The horizontal and vertical axes represent the mean ISI of 2-spike bursts and the mean spike number, respectively. The correlation was −0.15 for this cell. **(E)** Stimulus sequences preceding 2-spike bursts with an intraburst ISI of < 6.0 ms (top) and > 16.0 ms (bottom) are aligned with respect to the first spike in the bursts. **(F)** The thick black line indicates the average of all stimulus sequences preceding 2-spike bursts (2-BTA). The thick colored lines indicate the average of the stimulus sequences preceding 2-spike bursts with different intraburst ISIs. The thin lines indicate the SEM values calculated across bursts. **(G)** The thick yellow and blue lines indicate the deviations from 2-BTA, i.e., the average of the stimulus sequence preceding 2-spike bursts with the longest and shortest 50% of intraburst ISIs, respectively, from which the 2-BTA was subtracted. The thin lines indicate SEM values calculated across bursts. **(H)** Population analysis. The thick lines indicate the deviation from 2-BTA represented by the thick lines in (G) averaged among 19 cells that generated at least 1500 2-spike bursts. The thin lines represent SEM values calculated among the cells. **(I)** Analysis of $s_{D2,2\text{-spike}}$, the stimulus projected onto the deviation from the 2-BTA and collected at the time points of all 2-spike bursts and normalized to have a mean = 0 and a SD = 1. The dashed line shows $P(s_{D2,2\text{-spike}})$ averaged across the 19 cells. The thick solid lines are the probability distribution of $s_{D2,2\text{-spike}}$ at 2-spike bursts with short, medium, and long ISIs, $P(s_{D2,2\text{-spike}}|0-10\% \text{ ISI})$, $P(s_{D2,2\text{-spike}}|45-55\% \text{ ISI})$, and $P(s_{D2,2\text{-spike}}|90-100\% \text{ ISI})$, respectively, averaged across the 19 cells. The thin lines are the SEM calculated across the cells. **(J)** Similar to **(I)**, calculated for $s_{3-1,2\text{-spike}}$, the stimulus projected onto 3-BTA−1-BTA.

We projected the stimulus onto the deviation from the 2-BTA [27] and collected the values at all two-spike bursts, then normalized the values to have a mean = 0 and an SD = 1 ($s_{D2,2\text{-spike}}$, dashed line in Fig 3I). $s_{D2,2\text{-spike}}$ values were collected at the time points of 2-spike bursts with different ranges of ISIs (0–10, 45–55, and 90–100 percentile). The probability density distribution $P(s_{D2,2\text{-spike}}|0-10\% \text{ ISI})$, $P(s_{D2,2\text{-spike}}|45-55\% \text{ ISI})$, and $P(s_{D2,2\text{-spike}}|90-100\% \text{ ISI})$, were

shifted from each other according to the ISI range (mean = −0.56 ± 0.24, −0.03 ± 0.08, 0.66 ± 0.23, respectively, mean ± SD, $n$ = 19; Fig 3I), suggesting that the ISIs encode information regarding the stimulus sequence represented by the deviation from 2-BTA.

We also projected the stimulus onto 3-BTA−1-BTA and normalized the values collected at the time points of 2-spike bursts ($s_{3−1,2-spike}$). $s_{3−1,2-spike}$ showed little or no dependence on 2-spike burst ISIs (the mean of $P(s_{3−1,2-spike}|0−10\%$ ISI), $P(s_{3−1,2-spike}|45−55\%$ ISI), and $P(s_{3−1,2-spike}|90−100\%$ ISI) was 0.29 ± 0.20, −0.07 ± 0.06, 0.02 ± 0.24, respectively, mean ± SD, $n$ = 19; Fig 3J). This result suggests that 2-spike burst ISIs convey little to no information regarding the stimulus feature encoded by the spike number, which was consistent with the correlation analysis.

These results indicate that 2-spike burst ISIs encode the visual stimulus differently than the spike number. The encoded feature is an oscillatory sequence with a peak frequency of ~8.2 Hz and ON−OFF intervals of ~40 ms (Fig 3G and 3H). Thus, two-spike burst ISIs encode an oscillatory sequence that is much faster than the stimulus feature encoded by the spike number.

## The two ISIs of three-spike bursts encode the phase and amplitude of oscillatory components

We next investigated 3-spike burst ISIs. The first and second intraburst ISIs ($ISI_1$ and $ISI_2$) tended to be different for different events (Fig 4A and 4B) and carried information about the stimulus (4.6 ± 1.5 bits per burst, mean ± SD, $n$ = 41). In addition, the data suggest that $ISI_1$ and $ISI_2$ were modulated differently. For example, in Fig 4A, $ISI_1$ was similar between the first (red) and second (green) events but tended to be different for the third (blue) event. In contrast, $ISI_2$ was similar between the second (green) and third (blue) events, but different for the first event. Accordingly, in the two-dimensional plot of $ISI_1$ and $ISI_2$, bursts of different events occupied different locations, and events did not align one-dimensionally, but were located two-dimensionally (Fig 4B).

The above results suggest that the modulation of $ISI_1$ and $ISI_2$ has two degrees of freedom. The distribution of $ISI_1$ and $ISI_2$ had features that complicate further analysis. First, events with longer ISIs tended to have larger trial-to-trial ISI variability than events with shorter ISIs (Fig 4C). This inhomogeneous variability suggests that shorter ISIs represent information with a higher resolution than longer ISIs. Second, although $ISI_1$ and $ISI_2$ were modulated differently, they were not completely independent, but were correlated (Fig 4B; correlation coefficient = 0.16 ± 0.14, mean ± SD, $n$ = 41).

To correct the two features, we performed a coordinate transformation of the intraburst ISIs (see Materials and Methods). For the first feature, we calculated the mean and SD of the ISIs for each event and determined their dependence via linear fitting (Fig 4C). Variables $v_1$ and $v_2$ were defined as:

$$v_d = \log_{10}(\text{ISI}_d[\text{ms}] − m_d[\text{ms}]), \ d = 1, 2. \tag{1}$$

where $m_1$ and $m_2$ are the intersects shown in Fig 4C. The trial-to-trial variation of $v_1$ and $v_2$ in different events was more uniform than that of $ISI_1$ and $ISI_2$ (Fig 4D and Materials and Methods). To correct the second feature, $v_1$ and $v_2$ were linearly transformed as

$$v_d^* = c_{1d}v_d + c_{2d}, d = 1, 2$$

so that the probability distribution of $v_1^*$ and $v_2^*$ fits to the normal distribution with a mean = 0

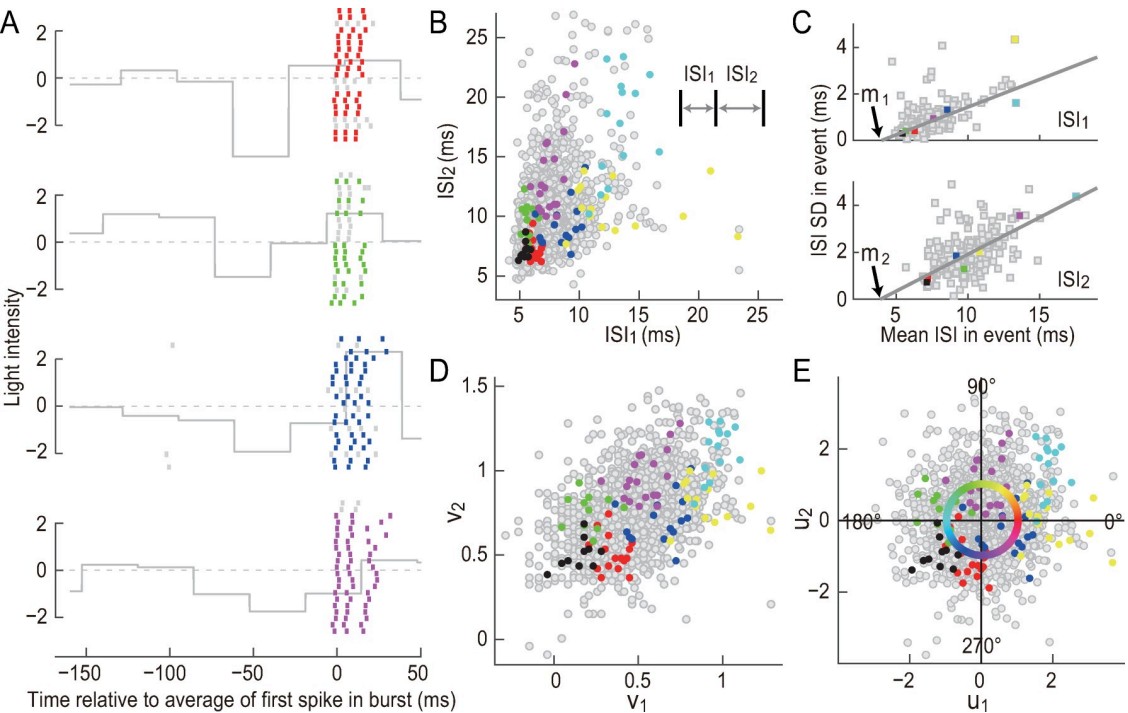

**Fig 4. Interspike interval (ISI) patterns of three-spike bursts.** Data from the cell shown in Fig 1. **(A)** Raster plot. The colored lines represent 3-spike bursts. The short gray lines are remaining spikes. The gray continuous line shows the normalized light intensity of the stimulus (the mean and SD are 0 and 1, respectively). **(B)** Distribution of $ISI_1$ and $ISI_2$. Each dot represents a 3-spike burst. The colored dots are bursts generated in seven different events, four of which are shown in **(A)** with the same color. The gray dots represent remaining bursts. **(C)** Variability of ISIs. Each dot represents an event. The horizontal and vertical axes are the average and SD of ISIs in an event, respectively. The gray lines show the linear fits. $m_1$ and $m_2$ are the intersects. **(D and E)** Coordinate transformation of intraburst ISIs. **(D)** $v_1$ and $v_2$ determined using Eq (1) provided in the text. Each dot represents a 3-spike burst. **(E)** $u_1$ and $u_2$ were determined by linear scaling of $v_1$ and $v_2$, as shown by Eq (2) provided in the text. The colors in the circle indicate the burst phases. ~1.5% of bursts are out of the range of the plot in (B), (D), and (E).

and a SD = 1, where $c_{1d}$ and $c_{2d}$ are constants. $v_1^*$ and $v_2^*$ were further linearly scaled as follows:

$$\begin{pmatrix} u_1 \\ u_2 \end{pmatrix} = \begin{pmatrix} p_1 & q_1 \\ p_2 & q_2 \end{pmatrix} \begin{pmatrix} c^{-1} & 0 \\ 0 & d^{-1} \end{pmatrix} \begin{pmatrix} p_1 & p_2 \\ q_1 & q_2 \end{pmatrix} \begin{pmatrix} v_1^* - \overline{v_1^*} \\ v_2^* - \overline{v_2^*} \end{pmatrix}, \qquad (2)$$

where $\overline{v_d^*}$ is the mean of $v_d^*$ $(d = 1, 2)$, $(p_1, p_2)$ and $(q_1, q_2)$ are the unit vectors that are parallel to the principle axes of the distribution of $(v_1^*, v_2^*)$, and $c^2$ and $d^2$ $(c>0, d>0)$ are the corresponding variances. $u_1$ and $u_2$ had an approximately circularly symmetric distribution with negligible correlation (Fig 4E; correlation coefficient = −0.01 ± 0.13, mean ± SD, n = 41). Thus, $u_1$ and $u_2$ were approximately independent, with different events occupying a similar amount of area (Fig 4E).

Using $u_1$ and $u_2$, we defined the "burst phase" for each burst (Fig 4E). Stimulus sequences preceding 3-spike bursts exhibited systematic differences according to the burst phase (Fig 5A). We subtracted the 3-BTA (Fig 5B) from the sequences preceding 3-spike bursts. The resulting deviations were oscillatory components with two or three ON peaks separated by 70–80 ms, with OFF peaks among them (Fig 5C–5E). The intervals between these peaks were almost constant, but their timing relative to the first spike of bursts shifted depending on the burst phase (Fig 5C–5E). When the burst phase increased from 0° to 360°, the peaks moved toward the first spikes of bursts (Fig 5C–5E), and the timing of the peaks showed

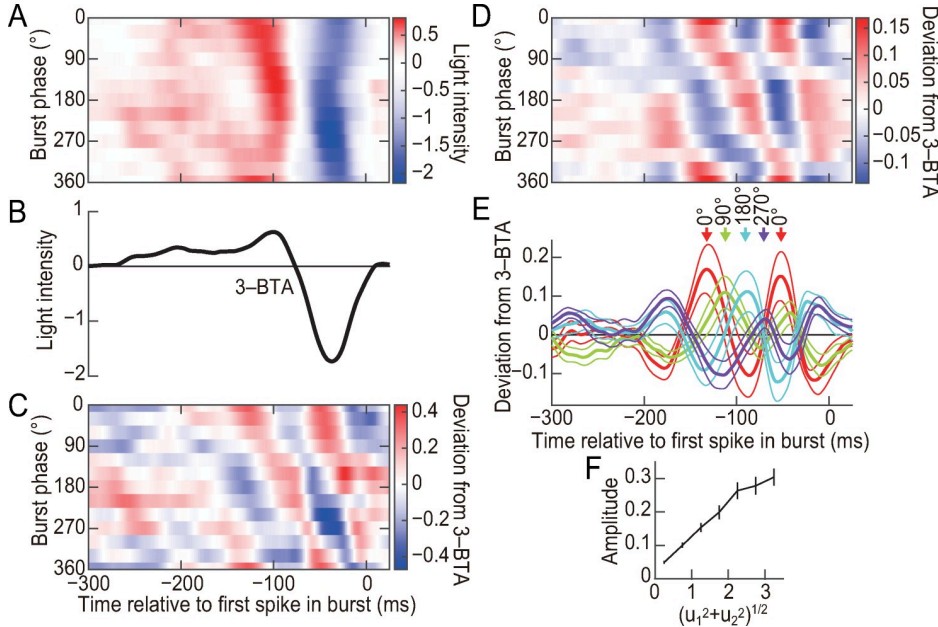

**Fig 5. 3-Spike burst ISI patterns encode the phase and amplitude of oscillatory components. (A–C)** Analysis of the cell shown in Fig 1. **(A)** Stimulus sequences preceding 3-spike bursts with different burst phases. Bursts were divided into 12 groups according to the burst phase (bin size = 30˚). Stimulus sequences preceding the bursts in each group were averaged and are shown in each row. **(B)** Average stimulus sequence preceding 3-spike bursts (3-BTA). **(C)** Deviation from 3-BTA calculated by subtracting the 3-BTA shown in (B) from the preceding sequences shown in (A). **(D–F)** Population analyses. **(D)** Same analysis as in (C), averaged among 19 cells that generated at least 1200 3-spike bursts. **(E)** Thick lines indicate data in (D) at the indicated burst phase. Thin lines indicate the SEM values calculated across the cells. Peaks around −100 ms are indicated. **(F)** Horizontal axis: $(u_1^2 + u_2^2)^{1/2}$, i.e., the distance of the point ($u_1$, $u_2$) from the origin in the $u_1$–$u_2$ plane in Fig 4E. Vertical axis: the root mean square of the oscillatory components between −200 and 25 ms. The error bars indicate SEM values for the 19 cells.

approximately linear dependence on the burst phase (Fig 5C–5E). These results indicate that the phase of 3-spike bursts encodes the temporal phase of an oscillatory component. In addition, we found that the distance of the point ($u_1$, $u_2$) from the origin of the $u_1$–$u_2$ plane encodes the amplitude of the oscillatory component (Fig 5F).

We also conducted similar analyses for bursts elicited with natural scene movies (Fig 6). The average stimulus sequences preceding bursts were slower than those observed for the spatially uniform stimulus (Fig 6B–6D). Nevertheless, 2-spike burst ISIs encoded oscillatory components (Fig 6C). Although the stimulus features encoded by 3-spike bursts with burst phases of 0˚ and 90˚ were similar (red and green in Fig 6D), those encoded by bursts of 180˚ and 270˚ (cyan and purple in Fig 6D, respectively) exhibited temporal shifts, similar to that observed for the spatially uniform stimulus.

In the analyses described above, stimulus sequences preceding bursts were aligned according to the first spike in bursts. We investigated whether the results depended on the stimulus alignment, by aligning the stimulus sequences on the second spike, the third spike, and the middle of the burst duration (i.e., the middle of the first and third spikes). The results obtained for the spatially uniform stimulus were similar to that obtained by aligning the sequences according to the first spike (S3A–S3I Fig and Fig 5). The encoding of the natural scene movie exhibited stronger dependence on the alignment (S4 Fig). In addition, because Eq 1 cannot define $v_1$ and $v_2$ for bursts with $ISI_1 \leq m_1$ or $ISI_2 \leq m_2$, these bursts were removed from the analyses described above. The incorporation of these bursts into the analysis with a small

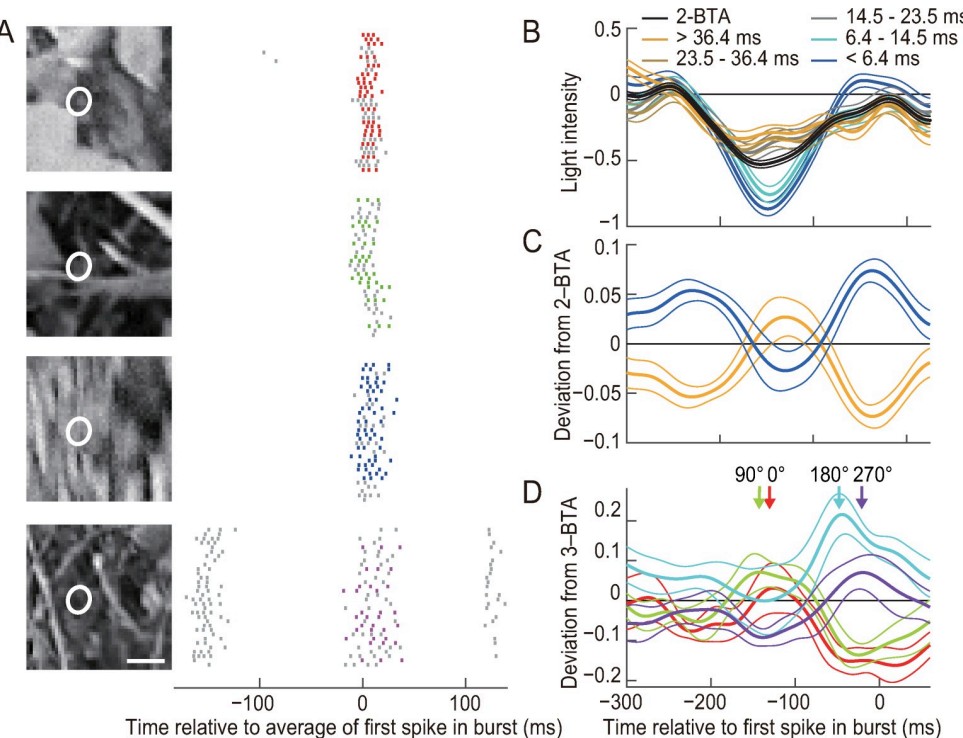

**Fig 6. ISI analysis of bursts elicited by natural scene stimulation. (A)** Responses of a single ganglion cell. (Right) Raster plots. The colored dots indicate 3-spike bursts. The gray dots represent remaining spikes. (Left) Image frames acquired at −60 ms relative to the average timing of the first spike in bursts. White ellipses show receptive field centers determined by reverse correlation and Gaussian fitting. Bar, 1 mm. **(B–D)** Analyses performed using light intensity at the receptive field center. **(B)** Analysis of 2-spike bursts of a single cell. The thick black line indicates the average of all stimulus sequences preceding 2-spike bursts (2-BTA). The thick colored lines indicate the average of stimulus sequences preceding 2-spike bursts with different intraburst ISIs. The thin lines indicate SEM values calculated across bursts. **(C)** Population analysis of the deviation from 2-BTA. For each cell, the deviation from 2-BTA was calculated by subtracting 2-BTA from the average stimulus sequence preceding 2-spike bursts with the longest and shortest 50% of intraburst ISIs. The thick lines indicate the deviation from 2-BTA averaged across the 13 cells that generated more than 800 2-spike bursts (yellow, longest ISIs; blue, shortest ISIs). The thin lines indicate the SEM values calculated across the cells. **(D)** Deviation from 3-BTA. For each cell, the deviation from 3-BTA was calculated by subtracting 3-BTA from the average stimulus sequence preceding 3-spike bursts with the burst phase within a specific range. The thick lines indicate the deviation from 3-BTA averaged across the 8 cells that generated more than 1000 3-spike bursts. The thin lines show SEM values calculated across the cells. The arrows indicate the first peak before the first spike in bursts.

modification of the method had little or no effect on the results (S3M–S3O Fig), suggesting that the encoding by bursts with very short ISIs is similar to that described above.

## Linear reconstruction fails to decode the burst pattern

We investigated whether 3-spike burst patterns encode stimulus sequences in a manner that can be decoded using the simple decoding algorithm, i.e., linear reconstruction. For this purpose, we calculated the spike-triggered averages (STAs) for spikes in 3-spike bursts. Linear reconstruction was performed by aligning three of the identical STAs on the three spikes of a burst and calculating the sum (Fig 7A and Materials and Methods). This reconstruction was carried out for all 3-spike bursts, and the 3-BTA and the deviation from the 3-BTA were calculated using the reconstructed stimuli. The reconstructed sequences were almost the same for all the burst phases (compare Figs 5A and 7B), and the deviation from the 3-BTA was much smaller than that of the actual data (compare Figs 5E and 7C). To quantify the amplitude of the deviation, the root mean square of the deviation from the 3-BTA was calculated for the

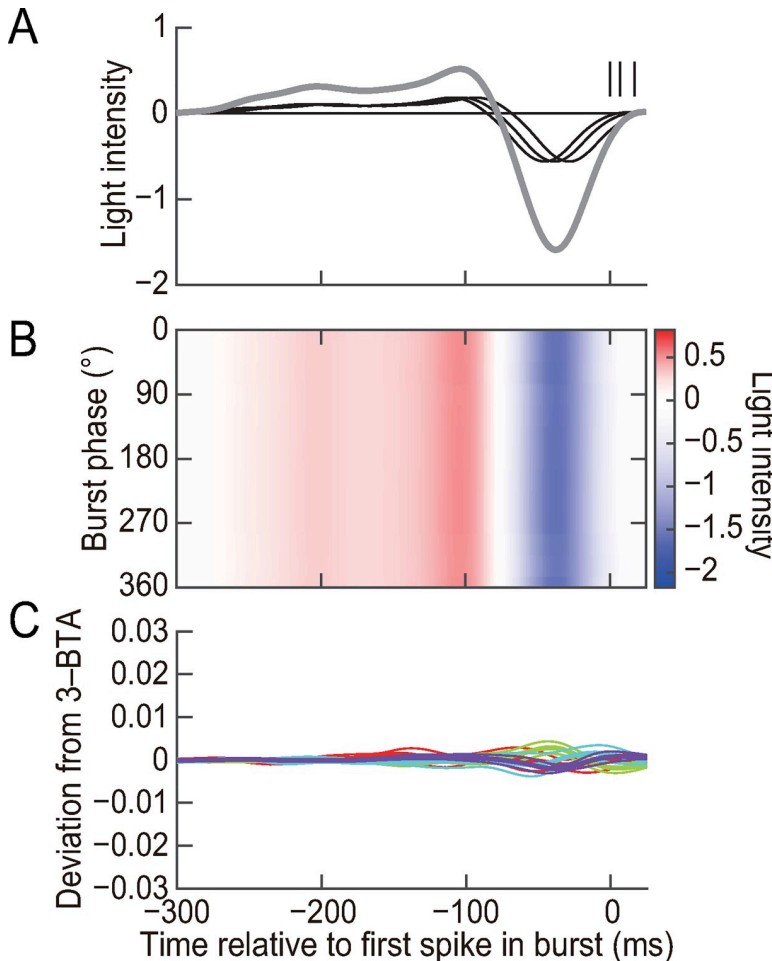

**Fig 7. Linear reconstruction of the stimulus using 3-spike bursts patterns. (A)** Short vertical lines represent spikes in an example burst where $ISI_1 = 7$ ms and $ISI_2 = 10$ ms. The three thin black lines indicate STAs calculated for 3-spike bursts of the cell used in (Fig 5A–5C) and aligned on the three spikes. The STAs were summed to calculate the reconstructed stimulus (thick gray line). **(B)** Analysis similar to (Fig 5A) conducted for stimuli reconstructed as in (A) using the same bursts used in (Fig 5A). Color-coding is the same as in (Fig 5A). **(C)** Population analysis similar to (Fig 5E), conducted for the reconstructed stimuli.

period between −200 and +25 ms and averaged for all burst phases. The values were significantly larger for the actual stimuli than for the reconstructed stimuli (0.107 ± 0.061 for the actual stimuli and 0.003 ± 0.002 for the reconstructed stimuli, mean ± SD, $n = 19$ cells that generated at least 1200 3-spike bursts, $P = 1.5 \times 10^{-7}$, two-tailed Mann–Whitney–Wilcoxon test). Thus, stimulus sequences encoded by the burst pattern cannot be decoded by the simple linear reconstruction.

## Two independent components of the burst patterns

The oscillatory component encoded by 2-spike burst ISIs had peak-to-peak intervals that are similar to those of the components encoded by 3-spike bursts (~80 ms; compare Figs 3H and 5E), suggesting that 2- and 3-spike burst ISIs are modulated by related stimulus features. To further characterize this similarity, we analyzed events in which both 2- and 3-spike bursts were generated (e.g., Fig 4A, the second top panel). For each of these events we calculated the average values of $u_1$ and $u_2$ for 3-spikes bursts and the average value of the 2-spike burst ISIs

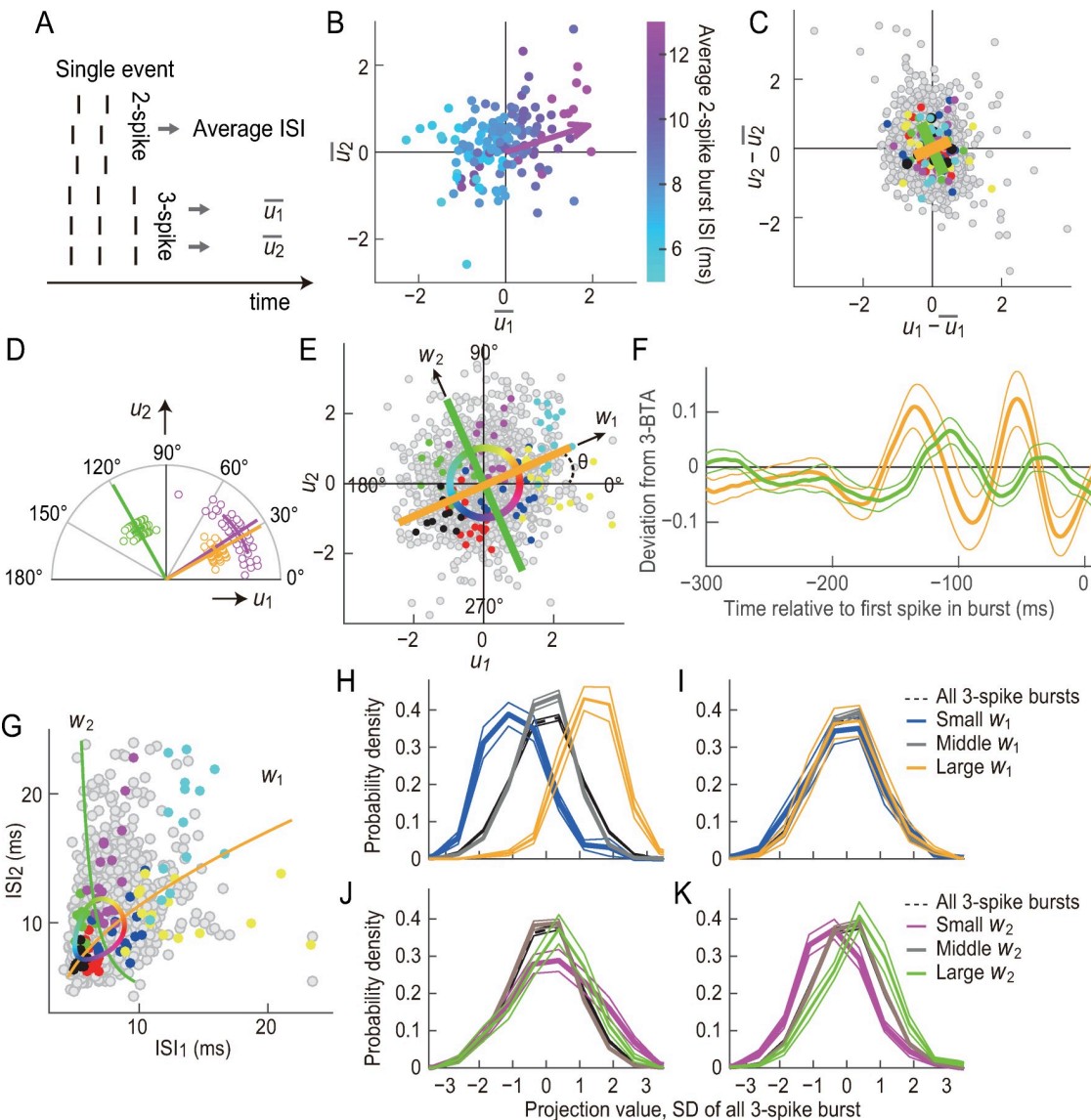

**Fig 8. Identification of independent components that determine burst patterns.** **(A)** The average value of 2-spike burst ISIs and the average values of 3-spike bursts $u_1$ and $u_2$ were determined for each event. **(B, C, E, and G)** Data from the cell shown in Fig 1. **(B)** Dependence of 2-spike burst ISIs on 3-spike burst patterns. Each dot represents an event in which the cell generated both 2- and 3-spike bursts. The horizontal and vertical axes are the averages of $u_1$ and $u_2$ of 3-spike bursts in an event, respectively. The color indicates the average ISI of 2-spike bursts. The arrow shows the direction of the steepest gradient of 2-spike burst ISIs. **(C)** Trial-to-trial variations of $u_1$ and $u_2$ in each event. Each dot represents a 3-spike burst. $\overline{u_1}$ and $\overline{u_2}$ represent the average $u_1$ and $u_2$ in each event. Dots with the same color are bursts in the same event. The yellow and green lines indicate the orientation of the principle components with the smaller and larger variances, respectively. The lengths represent the SD along the axes (compare with Fig 4E). **(D)** Population analyses for $n = 41$ cells. Each dot represents a cell. (Magenta) The direction of the steepest gradient of 2-spike burst ISIs, determined as in (B). (Yellow and green) Principle axes of the trial-to-trial variations of $u_1$ and $u_2$, determined as in (C). The yellow color shows the axis with the smaller variance. Straight and curved lines indicate the circular average and SD, respectively. **(E)** Distribution of $u_1$ and $u_2$. Each dot represents a 3-spike burst. Dots with the same color are bursts in the same event. The yellow and green lines are the principle axes shown in (C). $\theta$ represents the angle between the $u_1$ axis and the principle axis with the smaller variance (yellow line). The approximately independent components $w_1$ and $w_2$ are shown (compare with Fig 4E). **(F)** Stimulus sequences encoded by the independent components $w_1$ and $w_2$. For each cell, the deviation from 3-BTA were calculated by subtracting 3-BTA from the average stimulus sequence preceding 3-spike bursts with the burst phase within the range centered by $\theta$ (for $w_1$) and $\theta + 90°$ (for $w_2$). The thick lines indicate the deviation from 3-BTA averaged across 19 cells that generated at least 1200 3-spike bursts (yellow: $w_1$; green: $w_2$). The thin lines show SEM values calculated across the cells. **(G)** The axes and circle in (E) are plotted on the $ISI_1$–$ISI_2$ plane. **(H)** The stimulus was projected onto the stimulus feature encoded by $w_1$. $s_{w1,3\text{-spike}}$ was defined by collecting the values at the time points of all 3-spike bursts and normalizing them to have a mean = 0 and a SD = 1. The dashed line shows $P(s_{w1,3\text{-spike}})$ averaged for the 19 cells. The thin lines indicate the SEM calculated across the cells. The colored thick lines are

the probability distribution of $s_{w1,3\text{-spike}}$ at the time points of 3-spike bursts with a small, medium, and large $w_1$, i.e., $P(s_{w1,3\text{-spike}}|w_1<-2)$, $P(s_{w1,3\text{-spike}}|-0.5\leq w_1<0.5)$, and $P(s_{w1,3\text{-spike}}|2\leq w_1)$, respectively, averaged among the 19 cells. The thin lines show the SEM calculated across the cells. **(I)** Similar to (H), showing $P(s_{w2,3\text{-spike}})$, $P(s_{w2,3\text{-spike}}|w_1<-2)$, $P(s_{w2,3\text{-spike}}|-0.5\leq w_1<0.5)$, and $P(s_{w2,3\text{-spike}}|2\leq w_1)$. **(J)** Similar to (H), showing $P(s_{w1,3\text{-spike}})$, $P(s_{w1,3\text{-spike}}|w_2<-2)$, $P(s_{w1,3\text{-spike}}|-0.5\leq w_2<0.5)$, and $P(s_{w1,3\text{-spike}}|2\leq w_2)$. **(K)** Similar to (H), showing $P(s_{w2,3\text{-spike}})$, $P(s_{w2,3\text{-spike}}|w_2<-2)$, $P(s_{w2,3\text{-spike}}|-0.5\leq w_2<0.5)$, and $P(s_{w2,3\text{-spike}}|2\leq w_2)$. ~1.5% of bursts are out of the range of the plot in (C), (E), and (G).

(Fig 8A). Plotting the data on the $u_1-u_2$ plane showed that 2-spike burst ISIs differ systematically depending on the position of the events along the direction of ~30˚ (Fig 8B, arrow). The direction of the steepest gradient of 2-spike burst ISIs on the $u_1-u_2$ plane was 32.3˚ ± 15.6˚ (circular mean ± circular SD, $n = 41$; Fig 8D, magenta). This dependence suggests that 2-spike burst ISIs are modulated by an oscillatory component that modulates 3-spike burst patterns in the orientation of ~32.3˚ on the $u_1-u_2$ plane.

The above result raises the hypothesis that the retinal mechanism that modulates 2-spike ISIs also modulates 3-spike burst patterns in the orientation of ~32.3˚ on the $u_1-u_2$ plane. Since $u_1$ and $u_2$ are suggested to have two degrees of freedom, one possibility is that another independent mechanism modulates 3-spike burst patterns in the orthogonal orientation, i.e., ~122.3˚, on the $u_1-u_2$ plane. If such two independent mechanisms were present, modulation of $u_1$ and $u_2$ in the two orthogonal orientations would have independent trial-to-trial variations, and we tested this prediction. Although $u_1$ and $u_2$ had an approximately circularly symmetric distribution (Fig 4E), their trial-to-trial variations within each event had an asymmetric distribution (Fig 8C). The principal component analysis of this distribution indicated that the principal axes corresponding to the smaller and larger variances were in the orientations of $\theta = 28.6˚ \pm 7.7˚$ and $\theta + 90˚ = 118.6˚ \pm 7.7˚$ (circular mean ± circular SD, $n = 41$; yellow and green, respectively, in Fig 8C and 8D). This analysis indicates that modulations of $u_1$ and $u_2$ in these two orientations have approximately independent trial-to-trial variation. Because these axes (~28.6˚ and ~118.6˚, yellow and green, respectively, in Fig 8D) are close to those proposed for the hypothesis (~32.3˚ and ~122.3˚, the former is shown in magenta in Fig 8D), the results are in accordance with the presence of two independent mechanisms, one of which (~30˚) is common to 2- and 3-spike bursts.

We defined new variables, i.e., $w_1 = u_1 \cos\theta + u_2 \sin\theta$ and $w_2 = u_1 \cos(\theta+90˚) + u_2 \sin(\theta+90˚)$, which represent the approximately independent components (Fig 8E). The stimulus features encoded by these components (Fig 8F, and S2B and S2C Fig) were determined by calculating the deviation from 3-BTA for the phases $\theta$ and $\theta+90˚$, similarly to Fig 5E. The two encoded features were oscillatory sequences with a peak frequency of 9.4 ± 3.3 Hz for $\theta$ and 6.9 ± 4.8 Hz for $\theta+90˚$ (mean ± SD, $n = 19$ cells that generated at least 1200 3-spike bursts). The two sequences are approximately orthogonal to each other, i.e., components with ~1/4-cycle difference in phase (Fig 8F). As expected, the oscillatory component encoded by $w_1$ was similar to that encoded by 2-spike burst ISIs (compare Fig 8F, yellow with Fig 3H, yellow). The waveform of the sequences exhibited little to no dependence on whether the stimulus sequences was aligned on the first spike, the second spike, the third spikes, or the middle of the burst duration (Fig 8F and S3J–S3L Fig).

We projected the stimulus onto the stimulus features encoded by the two independent components $w_1$ and $w_2$ [27], and collected the values at the time points of all 3-spike bursts and normalized the values so that they have a mean = 0 and a SD = 0 ($s_{w1,3\text{-spike}}$ and $s_{w2,3\text{-spike}}$, dashed lines in Fig 8H–8K). We collected $s_{w1,3\text{-spike}}$ values at 3-spike bursts with $w_1<-2$, $-0.5\leq w_1<0.5$, and $2\leq w_1$. The result shows that $s_{w1,3\text{-spike}}$ depended on the $w_1$ value of 3-spike bursts (the mean of $P(s_{w1,3\text{-spike}}|w_1<-2)$, $P(s_{w1,3\text{-spike}}|-0.5\leq w_1<0.5)$, $P(s_{w1,3\text{-spike}}|2\leq w_1)$ was −0.88 ± 0.41, −0.04 ± 0.06, and 1.30 ± 0.38, respectively, mean ± SD, $n = 19$; Fig 8H). Moreover, the projection onto the stimulus feature encoded by $w_2$ exhibited a similar

dependence on the $w_2$ value of bursts (the mean of $P(s_{w2,3\text{-spike}}|w_2<-2)$, $P(s_{w2,3\text{-spike}}|-0.5 \leq w_2<0.5)$, and $P(s_{w2,3\text{-spike}}|2\leq w_2)$ was $-0.38 \pm 0.21$, $0.02 \pm 0.05$, and $0.31 \pm 0.38$, respectively, mean $\pm$ SD, $n = 19$; Fig 8K). These results confirm that $w_1$ and $w_2$ encode the information regarding the oscillatory stimuluses features.

In contrast, the projection onto the stimulus feature encoded by $w_2$ showed little or no dependence on the $w_1$ value (the mean of $P(s_{w2,3\text{-spike}}|w_1<-2)$, $P(s_{w2,3\text{-spike}}|-0.5\leq w_1<0.5)$, and $P(s_{w2,3\text{-spike}}|2\leq w_1)$ was $-0.11 \pm 0.35$, $0.02 \pm 0.06$, and $0.04 \pm 0.61$, respectively, mean $\pm$ SD, $n = 19$; Fig 8I), and vice versa (the mean of $P(s_{w1,3\text{-spike}}|w_2<-2)$, $P(s_{w1,3\text{-spike}}|-0.5\leq w_2<0.5)$, and $P(s_{w1,3\text{-spike}}|2\leq w_2)$ was $0.28 \pm 0.44$, $-0.07 \pm 0.09$, $0.18 \pm 0.40$, respectively, mean $\pm$ SD, $n = 19$; Fig 8J). These results suggest that $w_1$ and $w_2$ encode two largely independent stimulus features.

$w_1$ and $w_2$ corresponded to the components with smaller and larger trial-to-trial variation (Fig 8C and 8E). Given that the total distribution of $u_1$ and $u_2$ and, thus, that of $w_1$ and $w_2$ was circularly symmetric, $w_1$ was more informative than $w_2$. Consistently, the shift in the probability density distribution of the projection values was larger for $w_1$ than it was for $w_2$ (compare Fig 8H and 8K).

The two approximately independent components $w_1$ than $w_2$ encode oscillatory components that are approximately orthogonal to each other, i.e., oscillatory components with ~1/4 cycles difference in phase (Fig 8F). This result suggests that the two orthogonal oscillatory stimulus components modulate 3-spike burst patterns in the orthogonal orientations on the $u_1-u_2$ plane. This model is consistent with the above result that 3-spike burst patterns encode the amplitude and phase of an oscillatory component, since an oscillatory sequence with an arbitrary amplitude and phase can be approximated by a sum of two orthogonal oscillatory components with fixed phases and the same frequency.

## Discussion

Our results reveal that intraburst ISIs of retinal bursts encode oscillatory light intensity sequences that are much faster than the sequence encoded by the spike number; The spike number encodes a stimulus feature of ~4.7 Hz, while intraburst ISIs encode oscillatory sequences of ~6.9–9.4 Hz. When a burst has three spikes, the two intraburst ISIs combinatorially encode the amplitude and phase of the oscillatory component. These results therefore suggest that a $k$-spike burst ($k = 2, 3$) encodes $k$ different stimulus features by exploiting all the $k$ degrees of freedom, i.e., the spike number and $k-1$ ISIs. This simultaneous representation of multiple stimulus features enables multiplexed information coding, a mechanism that greatly increases the information transmission capacity [19, 28, 29].

### Mechanisms of the combinatorial ISI coding

Fig 9 shows a coding model that is consistent with our findings. The amplitude of slow light intensity modulation determines the spike number within a burst. Intraburst ISIs are regulated by two independent mechanisms that are driven by orthogonal fast oscillatory stimulus components, as suggested by the comparison between 2- and 3-spike bursts and the analysis of trial-to-trial variation. When a burst contains two spikes, the ISI is regulated by one of the two mechanisms, and thus 2-spike burst ISIs encode the amplitude of an oscillatory component of a fixed phase. When a burst has three spikes, the two mechanisms combinatorially determine the two ISIs. Because the two mechanisms are driven by the two orthogonal oscillatory components, the two ISIs of 3-spike bursts carry information about both the amplitude and phase of the oscillatory component. Modulation of the 3-spike ISI pattern by the common component is similar to modulation of the burst duration, i.e, $ISI_1 + ISI_2$ (see yellow in Fig 8G).

The two proposed mechanisms modulating intraburst ISIs exhibit approximately independent trial-to-trial variation, raising the possibility that the two mechanisms rely on two largely

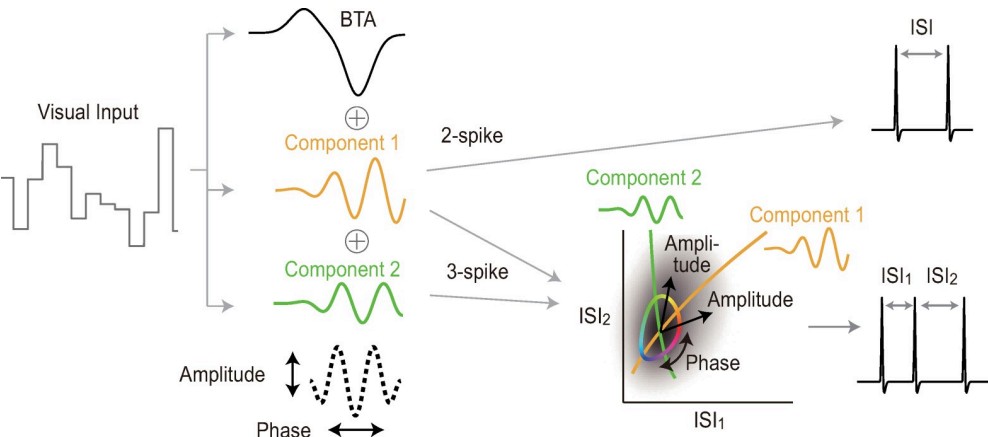

**Fig 9. Schematic view of burst coding.** The dotted line indicates the sum of the oscillatory components 1 and 2, whose amplitude and phase are encoded by the 3-spike burst pattern.

non-overlapping synaptic pathways. Such circuits, if present, may have different temporal properties, considering that the two mechanisms respond to two different temporal sequences. In the vertebrate retinae, bipolar cells have ~10 subtypes [30–33], and different subtypes have distinct physiological properties [33–35] and varying temporal response characteristics [36–38]. In addition, the inhibitory effect of amacrine cells on bipolar cells generates further variation of temporal properties [39]. Therefore, specific subsets of bipolar and amacrine cells may constitute largely non-overlapping synaptic pathways underling the combinatorial coding.

## Generality of the combinatorial ISI coding

Bursts with $\geq 4$ spikes were relatively rare in our experiment, which hampered the detailed analysis of those bursts. However, the amount of information about the stimulus encoded by the first and second ISIs in 4-spike bursts (4.6 ± 1.4 bits per burst, mean ± SD, $n = 41$) was similar to that observed for 3-spike bursts, suggesting that the ISIs of bursts with a large number of spikes also encode the stimulus. Further studies should clarify whether the combinatorial ISI coding occurs also for burst with four or more spikes.

Bursts elicited by the natural scene movie exhibited a combinatorial ISI coding that was similar to that observed for bursts elicited by the spatially uniform stimulus. The temporal features encoded by burst ISI patterns were slower than that observed for the spatially uniform stimulus. This difference might be a result of the slow temporal correlation of the natural scene movie (the temporal constant $\tau_{1/2}$ was 33 ms for the spatially uniform stimulus and 88 ms for the natural scene movie). The slow temporal correlation of the natural scene movie might also underlie the longer burst duration (3-spike burst duration: 14.4 ± 4.7 ms for the spatially uniform stimulus, $n = 41$; and 29.0 ± 9.8 ms for the natural scene movie, $n = 16$; mean ± SD).

When we aligned the stimulus sequences on the first spike, second spike, third spike, and middle of the burst duration of 3-spike bursts, the results were similar for the spatially uniform stimulus (S3 Fig). In contrast, the analysis of bursts elicited by the natural scene movie showed a greater dependence on the alignment (S4 Fig), presumably because of the longer burst duration. Because the alignment on the first spike in burst resulted in a similar phase dependence for both the spatially uniform stimulus and the natural scene movie (Figs 5 and 6), the brain may interpret the stimulus using the first spike as the temporal reference.

Although the present study investigated the coding of the temporal patterns of visual stimuli, retinal ISIs also depend on the spatial patterns [40]. Therefore, it is possible that burst

ISIs encode spatial information as well as temporal information, and that spatial information coding underlies the differences between the ISI coding of the spatially uniform stimulus and that of the natural scene movie.

## Information transmission to the brain

It is currently unclear to what extent the burst ISI information analyzed in this study is transmitted to the brain. However, in many brain regions, neuronal responses are sensitive to the millisecond-scale temporal structure of synaptic inputs [24]. For example, in synaptic transmission from the retina to the lateral geniculate nucleus (LGN), retinal spikes with ISIs of a few milliseconds are much more effective in eliciting LGN spikes than those with ISIs of > ~20 ms [41–47]. Consistently, LGN burst ISIs are sensitive to the millisecond-scale structure of current input [19]. Similar to synaptic connections from the retina to the LGN, those from the LGN to cortical neurons are more responsive to spikes with short ISIs than those with long ISIs [48]. Such dependence of neuronal responses on input ISIs suggests that bursts with different ISIs have different transmission efficiency. In addition, the dependence on ISIs varies among individual synaptic connections [46–48]. This variation suggests that individual synapses have different preference for bursts with different ISIs and thus may function as a system to decode the information conveyed by burst ISIs [24], which cannot be decoded by a simple linear algorithm.

## Conclusions

The present results suggest that the retina employs mechanisms to regulate multiple components of intraburst ISIs, and thereby encodes multiple stimulus features by exploiting all degrees of freedom of burst spike patterns, i.e., the spike number and multiple intraburst ISIs. This burst coding is likely to affect visual information transmission, as synaptic transmission is sensitive to ISIs. Because bursts occur in various regions of the brain, analyses similar to the present study may reveal previously overlooked information transmission in those regions.

## Materials and methods

### Ethics statement

All experiments were approved by the RIKEN Wako animal experiments committee (H16-2B023) and were performed according to the guidelines of the animal facilities of the RIKEN Center for Brain Science.

### Animals

Larval tiger salamanders were provided by Charles D. Sullivan Co. Inc., Nashville, Tennessee, USA.

### Recording and stimulation

Retinal recording was performed as described previously [49]. Dark-adapted retinae from larval male and female tiger salamanders were isolated in oxygenated Ringer's medium at 25˚C. A piece of the retina (2–4 mm in width) was mounted on a flat array of 61 microelectrodes (MED-P2H07A, Alpha MED Scientific Inc., Ibaraki, Osaka, Japan) and perfused with oxygenated Ringer's solution (2 mL/min; 25˚C). Spatially uniform white light (intensity refreshment at 30 Hz; mean and SD of the intensity were 4.0 and 1.4 mW/m$^2$, respectively) was projected through an objective lens using a CRT monitor (60-Hz refresh rate; E551, Dell Inc., Round Rock, Texas, USA) controlled by the Matlab Psychophysics Toolbox [50, 51] or a light-emitting diode (E1L53-AWOC2-01 5-B5, Toyoda Gosei, Japan). The light intensity sequence was a random Gaussian sequence (65.5–183.3 s). The same sequence was repeated typically more

than 20 times. For the natural scene stimulation, 200 s of a movie [52] was projected at 30 Hz using the CRT monitor ($64 \times 64$ pixels, 60.6 μm/pixel; the mean intensity was 4.0 mW/m$^2$). The temporal constant $\tau_{1/2}$, i.e., the half-width at half maximum of the autocorrelation of the spatially uniform stimulus and the natural scene movie was 33.3 ms and 87.8 ms, respectively. Amplified voltage signals from the electrodes were stored and action potentials of single units were isolated using a Matlab program (a gift from Dr. Stephan A. Baccus). Analyses were performed using stable cells with mean firing rates >1.7 Hz.

## Identification of bursts and events

Histograms of the ISIs were generated for each ganglion cell (S1A Fig). The histograms often had two distinct peaks representing shorter and longer ISIs, corresponding to intra- and inter-burst ISIs, respectively [3–5]. The threshold interval $T_{\text{thresh}}$ was set at the trough between the two peaks in the ISI histogram (S1A Fig). $T_{\text{thresh}}$ was 38.6 ± 20.0 ms (mean ± SD) for 41 cells stimulated with the spatially uniform stimulation, and 87.7 ± 18.5 ms (mean ± SD) for the 16 cells stimulated with the natural scene movie. If two consecutive spikes occurred with an interval shorter than $T_{\text{thresh}}$, they were incorporated into the same burst, while they were separated into two consecutive bursts if the interval was longer than $T_{\text{thresh}}$ (S1B Fig). The robustness of this method was examined as follows. Bursts were defined using various threshold intervals of ~10 ms to ~100 ms, and the rates of the isolated spikes and bursts with 2–7 spikes were measured (S1C Fig). $r_{-10}$, $r_0$, and $r_{+10}$ (Hz) denote the rates of the 2-spike bursts defined by the threshold intervals $T_{\text{thresh}}-10$ ms, $T_{\text{thresh}}$, and $T_{\text{thresh}}+10$ ms, respectively. The maximum rate change, $\max(|r_{-10}-r_0|,|r_{+10}-r_0|)/r_0$, was 0.021 ± 0.020 (mean ± SD, $n = 41$) for the cells stimulated with the spatially uniform stimulation, and 0.022 ± 0.014 (mean ± SD, $n = 16$) for the cells stimulated with the natural scene movie. These small values indicate the robustness of the method. The median intraburst ISIs of the 2-spike bursts and the median of the duration ($\text{ISI}_1 + \text{ISI}_2$) of the 3-spike bursts were 8.1 ± 2.8 and 14.4 ± 4.7 ms (mean ± SD, $n = 41$), respectively, for the cells stimulated with the spatially uniform stimulus, and 15.4 ± 5.6 and 29.0 ± 9.8 ms (mean ± SD, $n = 16$), respectively, for the cells stimulated with the natural scene movie.

Events were determined as follows. The first spikes of bursts were determined in all stimulus repeats (black lines in S1D Fig, top). Let the timing of the first spikes in $h$-th repeat be $(t_1^{(h)}, t_2^{(h)}, t_3^{(h)}, \cdots)$, where $1 \le h \le N_{Rep}$ and $N_{Rep}$ is the number of repeats; and $0 \le t_g^{(h)} \le T$ for all $h$ and the burst number $g$, where $T$ is the length of one repeat of the stimulus. The first spikes of all stimulus repeats are:

$$(t_1^{(1)}, t_2^{(1)}, t_3^{(1)}, \cdots)$$

$$(t_1^{(2)}, t_2^{(2)}, t_3^{(2)}, \cdots)$$

$$\vdots$$

$$(t_1^{(h)}, t_2^{(h)}, t_3^{(h)}, \cdots)$$

$$\vdots$$

$$(t_1^{(Nrep)}, t_2^{(Nrep)}, t_3^{(Nrep)}, \cdots)$$

A new time sequence, the merged train of first spikes, was constructed by collecting and sorting all of the first spikes in the above sequences (S1D Fig, middle):

$$(t_1^{(2)}, t_1^{(1)}, \cdots, t_1^{(Nrep)}, \cdots, t_1^{(h)}, \cdots, t_2^{(h)}, \cdots, t_2^{(Nrep)}, t_2^{(1)}, \cdots, t_2^{(2)}, \cdots, t_3^{(2)}, \cdots, t_3^{(Nrep)}, t_3^{(h)}, \cdots, t_3^{(1)}, \cdots)$$

Note that the exact order depended on the data. The intervals between these first spikes in the merged train were determined and their histogram was generated (S1E Fig). The histogram had two peaks, suggesting that the first spikes, when merged across stimulus repeats, tend to form discrete clusters (S1D Fig, middle). Because this clustering indicates that bursts tended to occur with a similar timing across different repeats of the stimulus, each cluster defined an event. Because $T_{thresh}$ was located at the trough of the two peaks of the histogram (S1D Fig), it was used to determine clusters (and thus events); if the two first spikes in the merged train were closer than $T_{thresh}$, they were incorporated into the same event; otherwise, they were assigned into two consecutive events (S1D Fig, middle and bottom). In rare cases, bursts occurred with a large timing jitter in different repeats, and two consecutive bursts in one repeat were incorporated into one event (S1F Fig). In this case, the two consecutive bursts in the same event were treated as a single burst, which had an exceptionally long intraburst ISI $\geq T_{thresh}$. However, these exceptional bursts represented only 2.7% ± 2.7% of all bursts in the cells that were stimulated with the spatially uniform stimulation (mean ± SD, $n = 41$), and 3.5% ± 1.1% for the cells that were stimulated with the natural scene movie (mean ± SD, $n = 16$). These small fractions indicate that the exceptional cases were rare and that the event identification mostly assigned at most one burst per repeat into one event.

## Experimental design and statistical analyses

The number of analyzed cells and retinas was as follows; $n = 41$ cells in 15 retinas for the spatially uniform stimulation, and $n = 16$ cells in 4 retinas for the natural scene stimulation. Ganglion cells that generated only a small number of 2- or 3-spike bursts were removed from the analyses of 2- or 3-spike bursts, respectively (see legends of Figs 3H–3J, 5D, 6C, 6D, 8F, 8H–8K, and S2, S3, and S4 Figs).

**Mutual information.** The mutual information about the stimulus encoded by the spike number was obtained as follows [1]. Let $T$ be the length of one repeat of the stimulus. Before receiving spikes, all stimulus time points $0 \leq t \leq T$ were equally probable for the receiver; thus, the prior probability distribution of the stimulus was $P(t) = \frac{1}{T}$, and the prior entropy of the stimulus was $S[P(t)] = \int_0^T \frac{1}{T} \log_2 \frac{1}{T} dt = \log_2 T$, where $S[\cdot]$ denotes entropy. Let $r_k(t)$ be the rate of $k$-spike bursts in response to the repeated stimulation. The rates were determined using the first spikes in bursts, and 1-spike bursts represent isolated spikes. By denoting the average $\overline{r_k} = \frac{1}{T} \int_0^T r_k(t) dt$, the conditional probability distribution of the stimulus after receiving a $k$-spike bursts was $P(t|k\text{-spike}) = \frac{r_k(t)}{\overline{r_k} T}$. Thus, the information about the stimulus encoded per burst after the discrimination of spike numbers was:

$$I_{burst,number} = S[P(t)] - \sum_k P(k\text{-spike}) S[P(t|k\text{-spike})]$$

$$= \log_2 T - \sum_k P(k\text{-spike}) \int_0^T \frac{r_k(t)}{\overline{r_k} T} \log_2 \frac{r_k(t)}{\overline{r_k} T} dt$$

where $P(k\text{-spike})$ is the probability that a burst has $k$ spikes. The rate of all bursts was $r_{burst}(t) = \sum_k r_k(t)$, and its average was $\overline{r_{burst}} = \sum_k \overline{r_k}$. The conditional probability distribution of the stimulus after receiving a burst was $P(t|burst) = \frac{r_{burst}(t)}{\overline{r_{burst}} T}$. Thus, the information about the

stimulus encoded per burst when spike numbers were not discriminated was:

$$I_{\text{burst}} = S[P(t)] - S[P(t|\text{burst})] = \log_2 T - \int_0^T \frac{r_{\text{burst}}(t)}{r_{\text{burst}}T} \log_2 \frac{r_{\text{burst}}(t)}{r_{\text{burst}}T} \, dt$$

The information about the stimulus encoded by the spike number in bursts was the difference between the two types of information:

$$I_{\text{number}} = I_{\text{burst,number}} - I_{\text{burst}} = S[P(t|\text{burst})] - \sum_k P(k\text{-spike}) S[P(t|k\text{-spike})].$$

This notation indicates that $I_{\text{number}}$ indicates how far the stimulus entropy can be reduced by knowing the spike number after receiving a burst.

To calculate the information encoded by 2-spike burst ISIs, ISIs were divided into temporal bins with a width of $\Delta t$. Similar to above, two information values were defined:

$$I_{\text{2-spike,ISI}}(\Delta t) = S[P(t)] - \sum_{l=0} P(l\Delta t \leq \text{ISI} < (l+1)\Delta t) S[P(t|\text{2-spike}, l\Delta t \leq \text{ISI} < (l+1)\Delta t)]$$

and

$$I_{\text{2-spike}} = S[P(t)] - S[P(t|\text{2-spike})],$$

where $P(l\Delta t \leq \text{ISI} < (l+1)\Delta t)$ is the probability that a 2-spike burst has an ISI within the range $l\Delta t \leq \text{ISI} < (l+1)\Delta t$, and $l = 0, 1, 2, \cdots$. The difference of these two values,

$$I_{\text{ISI}}(\Delta t) = I_{\text{2-spike,ISI}}(\Delta t) - I_{\text{2-spike}}$$

$$= S[P(t|\text{2-spike})] - \sum_{l=0} P(l\Delta t \leq \text{ISI} < (l+1)\Delta t) S[P(t|\text{2-spike}, l\Delta t \leq \text{ISI} < (l+1)\Delta t)]$$

represents the information encoded by 2-spike burst ISIs with a resolution of $\Delta t$. The actual information about the stimulus encoded by the ISI of 2-spike bursts is the limit $I_{\text{ISI}} = \lim_{\Delta t \to 0} I_{\text{ISI}}(\Delta t)$. To estimate this limit, $I_{\text{ISI}}(\Delta t)$ was calculated using various $\Delta t$ values. Because $I_{\text{ISI}}(\Delta t)$ showed a linear dependence on $\Delta t$ in the range of $\Delta t = 1.5$–4 ms, $I_{\text{ISI}}(\Delta t)$ values were linearly fitted within this range, and the limit $\Delta t \to 0$ was estimated using the fit. The information encoded by 3-spike ISIs was similarly calculated by dividing $\text{ISI}_1$ and $\text{ISI}_2$ into bins and estimating the limit.

**Burst-triggered average (BTA).** For each cell, the 1-BTA (Fig 2A) was calculated as follows. Let $t_1, t_2, \cdots, t_N$ be the time of isolated spikes of the $k$-th cell, where $N$ is the number of isolated spikes recorded from the cell. Let $St(t)$ be the light intensity at time $t$ (mean = 0, SD = 1). 1-BTA for the $k$-th cell, $1\text{-BTA}_k(\tau)(-300 \text{ ms} \leq \tau \leq 0 \text{ ms})$, was determined by

$$1\text{-BTA}_k(\tau) = \mathcal{F}_{16} * \left\{ \frac{1}{N} \sum_{i=1}^N St(t_i + \tau) \right\},$$

where $\mathcal{F}_{16}$ is a smoothing filter that is defined by

$$\mathcal{F}_{16} * g(\tau) = \frac{1}{16 \text{ ms}} \int_{-8 \text{ ms}}^{+8 \text{ ms}} g(\tau + t) dt \qquad (3)$$

and performs temporal averaging over 16 ms to remove high-frequency noise. $2\text{-BTA}_k(\tau)$, $3\text{-BTA}_k(\tau)$, and $4\text{-BTA}_k(\tau)$ were similarly calculated using the time of the first spikes of 2-, 3-, and 4-spike bursts.

Let $N_{\text{cells}}$ be the number of cells. The average 1-BTA was:

$$1\text{-BTA}_{\text{ave}}(\tau) = \frac{1}{N_{\text{cells}}} \sum_{k=1}^{N_{\text{cells}}} 1\text{-BTA}_k(\tau).$$

The SEM among cells was:

$$1\text{-BTA}_{\text{SEM}}(\tau) = \frac{1}{\sqrt{N_{\text{cells}}}} \left[ \frac{1}{N_{\text{cells}}} \sum_{k=1}^{N_{\text{cells}}} \{1\text{-BTA}_k(\tau) - 1\text{-BTA}_{\text{ave}}(\tau)\}^2 \right]^{\frac{1}{2}}.$$

The average and SEM were similarly calculated for 2-, 3-, and 4-spike bursts and are shown in Fig 2A.

In Fig 2B, the average 3-BTA−1-BTA was 3-BTA$_{\text{ave}}(\tau)$−1-BTA$_{\text{ave}}(\tau)$. The SEM was:

$$\frac{1}{\sqrt{N_{\text{cells}}}} \left[ \frac{1}{N_{\text{cells}}} \sum_{k=1}^{N_{\text{cells}}} \{(3\text{-BTA}_k(\tau) - 1\text{-BTA}_k(\tau)) - (3\text{-BTA}_{\text{ave}}(\tau) - 1\text{-BTA}_{\text{ave}}(\tau))\}^2 \right]^{\frac{1}{2}}.$$

**Projection onto 3-BTA−1-BTA.** The projections shown in Fig 2C and 2D were analyzed as follows [27]. Let $St(t)$ be the light intensity at time $t$ (mean = 0 and SD = 1) and 1-BTA$_k(\tau)$ and 3-BTA$_k(\tau)$ be the 1-BTA and 3-BTA of the $k$-th cell. The raw projection values onto 3-BTA−1-BTA were:

$$s^*_{3-1,k}(t) = \frac{1}{300 \text{ ms}} \int_{-300 \text{ ms}}^{0 \text{ ms}} St(t + \tau)\{3\text{-BTA}_k(\tau) - 1\text{-BTA}_k(\tau)\}d\tau.$$

The values were normalized as follows:

$$s_{3-1,k}(t) = \{s^*_{3-1,k}(t) - \text{av}\}/\text{sd}$$

where av and sd are the average and standard deviation of $s^*_{3-1,k}(t)$. The probability density distribution $P(s_{3-1,k})$ was calculated for all cells ($k = 1, 2, \cdots, N_{\text{cells}}$, where $N_{\text{cells}}$ is the number of cells). The mean:

$$P(s_{3-1}) = \frac{1}{N_{\text{cells}}} \sum_{k=1}^{N_{\text{cells}}} P(s_{3-1,k})$$

and SEM:

$$\frac{1}{\sqrt{N_{\text{cells}}}} \left[ \frac{1}{N_{\text{cells}}} \sum_{k=1}^{N_{\text{cells}}} \{P(s_{3-1,k}) - P(s_{3-1})\}^2 \right]^{\frac{1}{2}}$$

were calculated, and the mean $P(s_{3-1})$ is presented as a dashed line in Fig 2C. The SEM was not shown in Fig 2C because it was very small. Let $t_1^{(1)}, t_2^{(1)}, \cdots, t_{N_1}^{(1)}$ be the time of isolated spikes of the $k$-th cell, where $N_1$ is the number of isolated spikes recorded from the cell. The projected values at the isolated spikes, i.e., $s_{3-1,k}(t_1^{(1)}), s_{3-1,k}(t_2^{(1)}), \cdots, s_{3-1,k}(t_{N_1}^{(1)})$, were collected and their probability density distribution $P(s_{3-1,k}|1\text{-spike})$ was calculated. $P(s_{3-1,k}|1\text{-spike})$ was determined for all cell $k$ ($k = 1 \cdots N_{\text{cells}}$), and the average $P(s_{3-1}|1\text{-spike})$ is shown as the thickest line in Fig 2C. The SEM is also shown. Similar analyses were performed for 2-spike, 3-spike, and 4-spike $P(s_{3-1}|2\text{-spike})$, $P(s_{3-1}|3\text{-spike})$, and $P(s_{3-1}|4\text{-spike})$ in Fig 2C.

For cell $k$, let $t_1, t_2, \cdots, t_{N_{burst}}$ be the time of isolated spikes and the first spikes of bursts, where $N_{\text{burst}}$ is the number of isolated spikes and bursts recorded from the cell. The average and standard deviation of $s_{3-1,k}^*(t)$ values at the isolated spikes and bursts,

$\text{av}_{\text{burst}} = \frac{1}{N_{\text{burst}}} \sum_{i=1}^{N_{\text{butst}}} s_{3-1,k}^*(t_i)$, and $\text{sd}_{\text{burst}} = \left[ \frac{1}{N_{\text{burst}}} \sum_{i=1}^{N_{\text{butst}}} \left\{ s_{3-1,k}^*(t_i) - \text{av}_{\text{burst}} \right\}^2 \right]^{\frac{1}{2}}$, were determined. $s_{3-1,k}^*(t)$ value at the isolated spikes and bursts was normalized to define

$s_{3-1,\text{burst},k}(i) = \left\{ s_{3-1,k}^*(t_i) - \text{av}_{\text{burst}} \right\} / \text{sd}_{\text{burst}}$ $(i = 1, 2, \cdots, N_{\text{burst}})$. $P(s_{3-1,\text{burst},k})$, the probability density of $s_{3-1,\text{burst},k}(i)$, was calculated for all cell $k$ $(k = 1 \cdots N_{\text{cells}})$, and the average $P(s_{3-1,\text{burst}})$ is shown as a dashed line in Fig 2D. The SEM is also shown. For cell $k$, $s_{3-1,\text{burst},k}(i)$ values at the time of isolated spikes were collected (note that $s_{3-1,\text{burst},k}(i)$ is defined for all isolated spikes and bursts), and the probability density distribution $P(s_{3-1,\text{burst},k}|1\text{-spike})$ was defined. $P(s_{3-1,\text{burst},k}|1\text{-spike})$ was calculated for all cell $k$ and the average $P(s_{3-1,\text{burst}}|1\text{-spike})$ is shown as the thickest line in Fig 2D. The SEM is also shown. $P(s_{3-1,\text{burst}}|2\text{-spike})$, $P(s_{3-1,\text{burst}}|3\text{-spike})$, and $P(s_{3-1,\text{burst}}|4\text{-spike})$, were similarly calculated and shown in Fig 2D.

**Correlation between the spike number and the intraburst ISIs.** The correlations between the spike number and the intraburst ISIs of the bursts were investigated using data from ganglion cells that were stimulated with the spatially uniform stimulation, as follows. For each event $j$ in which at least one 2-spike burst occurred, the average number of spikes ($n^{(j)}$) and the average intraburst ISIs of the 2-spike bursts ($m^{(j)}$) were determined. Fig 3D shows the distribution of ($m^{(j)}, n^{(j)}$) for one cell. The correlation between $n^{(j)}$ and $m^{(j)}$ was then calculated across events, as follows:

$$C = \frac{\frac{1}{N_{\text{Event2}}} \sum_j \left\{ (n^{(j)} - \overline{n})(m^{(j)} - \overline{m}) \right\}}{\sqrt{\frac{1}{N_{\text{Event2}}} \sum_j (n^{(j)} - \overline{n})^2} \sqrt{\frac{1}{N_{\text{Event2}}} \sum_j (m^{(j)} - \overline{m})^2}},$$

where $N_{\text{Event2}}$ denotes the number of events in which at least one 2-spike burst occurred, the sum $\sum_j(\cdot)$ is calculated for these events, and $\overline{n}$ and $\overline{m}$ denote the average of $n^{(j)}$ and $m^{(j)}$ among those events, respectively.

**Deviation from 2-BTA.** The deviation from 2-BTA was calculated for each cell as follows. The 2-BTA of the $k$-th cell (2-BTA$_k(\tau)$) was calculated as described above. Let $N_2$ be the number of 2-spike bursts recorded from the cell and $t_i$ $(i = 1, \cdots, N_2)$ be the time of the first spike of the $i$-th 2-spike burst. Let $A_{\text{L},k}$ be the set of index $i$ for which the $i$-th 2-spike bursts has intraburst ISIs longer than ISI$_{\text{median}}$, where ISI$_{\text{median}}$ is the median of intraburst ISIs of 2-spike bursts recorded from the cell. Thus,

$$A_{\text{L},k} = \{i | \text{intraburst ISI of } i\text{-th 2-spike burst of } k\text{-th cell} > \text{ISI}_{\text{median}} \}.$$

The deviation from 2-BTA for bursts with the longest 50% of intraburst ISIs, $\text{Dev}_{\text{L},k}(\tau)$, was defined by the following calculation:

$$\text{Dev}_{\text{L},k}(\tau) = \mathcal{F}_{16} * \left\{ \frac{1}{A_{\text{L},k}} \sum_{i \in A_{\text{L},k}} St(t_i + \tau) \right\} - 2\text{-BTA}_k(\tau),$$

where $\mathcal{F}_{16}$ is the smoothing filter defined by Eq 3, #$A_{\text{L},k}$ is the number of indices in $A_{\text{L},k}$ and $St(t)$ is the light intensity at time $t$.

In Fig 3G, the SEM was calculated among bursts:

$$\text{DevSEM}_{\text{L},k}(\tau) = \frac{1}{\sqrt{\#A_{\text{L},k}}}\left[\frac{1}{\#A_{\text{L},k}}\sum_{i\in A_{\text{L},k}}\{(\mathcal{F}_{16}*St(t_i+\tau)-2\text{-BTA}_k(\tau))-\text{Dev}_{\text{L},k}(\tau)\}^2\right]^{\frac{1}{2}}$$

$\text{Dev}_{\text{S},k}(\tau)$ and $\text{DevSEM}_{\text{S},k}(\tau)$ were similarly determined for 2-spike bursts with the shortest 50% of intraburst ISIs. Fig 3G shows $\text{Dev}_{\text{L},k}(\tau)$, $\text{DevSEM}_{\text{L},k}(\tau)$, $\text{Dev}_{\text{S},k}(\tau)$, and $\text{DevSEM}_{\text{S},k}(\tau)$, which were calculated for the cell used in Fig 3A.

The deviation for bursts with the longest ISIs averaged across cells was:

$$\text{Dev}_{\text{L,ave}}(\tau) = \frac{1}{N_{\text{cells}}}\sum_{k=1}^{N_{\text{cells}}}\text{Dev}_{\text{L},k}(\tau)$$

$$\text{Dev}_{\text{L,SEM}}(\tau) = \frac{1}{\sqrt{N_{\text{cells}}}}\left[\frac{1}{N_{\text{cells}}}\sum_{k=1}^{N_{\text{cells}}}\{\text{Dev}_{\text{L},k}(\tau)-\text{Dev}_{\text{L,ave}}(\tau)\}^2\right]^{\frac{1}{2}},$$

where $N_{\text{cells}}$ is the number of cells. $\text{Dev}_{\text{S,ave}}(\tau)$ and $\text{Dev}_{\text{S,SEM}}(\tau)$ were similarly calculated for bursts with the shortest 50% of intraburst ISIs. $\text{Dev}_{\text{L,ave}}(\tau)$, $\text{Dev}_{\text{L,SEM}}(\tau)$, $\text{Dev}_{\text{S,ave}}(\tau)$, and $\text{Dev}_{\text{S,SEM}}(\tau)$ are shown in Fig 3H.

The raw projection value for the $k$-th cell is [27]:

$$s_{\text{D2},k}^*(t) = \frac{1}{300\text{ ms}}\int_{-300\text{ ms}}^{0\text{ ms}}St(t+\tau)\{\text{Dev}_{\text{L},k}(\tau)-\text{Dev}_{\text{S},k}(\tau)\}d\tau.$$

The average and standard deviation of $s_{\text{D2},k}^*(t)$ values at 2-spike bursts, $\text{av}_{\text{2-spike}} = \frac{1}{N_2}\sum_{i=1}^{N_2}s_{\text{D2},k}^*(t_i)$ and $\text{sd}_{\text{2-spike}} = \left[\frac{1}{N_2}\sum_{i=1}^{N_2}\{s_{\text{D2},k}^*(t_i)-\text{av}_{\text{2-spike}}\}^2\right]^{\frac{1}{2}}$ were determined and the normalized value $s_{\text{D2,burst},k}(i) = \{s_{\text{D2},k}^*(t_i)-\text{av}_{\text{2-spike}}\}/\text{sd}_{\text{2-spike}}$ was defined for all 2-spike burst $i$ ($i$ =1, 2, $\cdots$, $N_2$). $P(s_{\text{D2,burst},k})$, the probability density of $s_{\text{D2,burst},k}(i)$, was calculated for all cell $k$ ($k = 1\cdots N_{\text{cells}}$), and the average $P(s_{\text{D2,burst}})$ is shown as a dashed line in Fig 3I. The SEM is also shown. $s_{\text{D2,burst},k}(i)$ values for 2-spike bursts with intraburst ISIs of 0–10 percentile, 45–55 percentile, and 90–100 percentile were collected (note that $s_{\text{D2,burst},k}(i)$ is defined for all 2-spike burst $i$). The probability density distributions of these values, $P(s_{\text{D2,2-spike},k}|0-10\%\text{ ISI})$, $P(s_{\text{D2,2-spike},k}|45\%-55\%\text{ ISI})$, and $P(s_{\text{D2,2-spike},k}|90-100\%\text{ ISI})$, were averaged across all cells and shown in Fig 3I. In addition, the projection onto 3-BTA$_k(\tau)$−1-BTA$_k(\tau)$ was calculated. These values were collected at 2-spike bursts and normalized ($s_{3-1,\text{2-spike}}$). $s_{3-1,\text{2-spike}}$ was processed similarly to $s_{\text{D2,burst},k}$ (Fig 3J).

**Coordinate transformation of 3-spike bursts.** Coordinate transformation of 3-spike burst ISIs was performed for each cell as follows (Fig 4B–4D). For 3-spike bursts in event $j$, $\overline{\text{ISI}}_1^{(j)}$ and $\text{SD}_1^{(j)}$ represent the average and standard deviation of ISI$_1$, respectively. Event $j$ with a larger $\overline{\text{ISI}}_1^{(j)}$ tended to have a larger $\text{SD}_1^{(j)}$ (Fig 4C), indicating the inhomogeneous variability of ISIs. To reduce this inhomogeneity, the following coordinate transformation was performed. $\text{SD}_1^{(j)}$ was linearly fitted with $\overline{\text{ISI}}_1^{(j)}$ ($j = 1,\cdots,N_{\text{Ev}}$; where $N_{\text{Ev}}$ is the number of events with 3-spike bursts) as $\text{SD}_1^{(j)} \cong a_1(\overline{\text{ISI}}_1^{(j)} - m_1)$, where $a_1$ and $m_1$ are constants (Fig 4C, top). Using the $m_1$, a variable, $v_1 = \log_{10}(\text{ISI}_1[\text{ms}]-m_1[\text{ms}])$, was defined. ISI$_1$ and $v_1$ had the following relationship: $\frac{dv_1}{d\text{ISI}_1} = (\text{ISI}_1 - m_1)^{-1}(\log 10)^{-1}$. If ISI$_1$ and ISI$_1$+$\Delta$ISI$_1$ ($\Delta$ISI$_1$ is small) are transformed into $v_1$ and $v_1$+$\Delta v_1$, respectively, then $\frac{\Delta v_1}{\Delta\text{ISI}_1} \cong (\text{ISI}_1 - m_1)^{-1}(\log 10)^{-1}$; thus,

$\Delta v_1 \cong \frac{\Delta ISI_1}{ISI_1 - m_1} (\log 10)^{-1}$. Therefore, when $\overline{ISI}_1^{(j)}$ and $\overline{ISI}_1^{(j)} \pm SD_1^{(j)}$ are transformed into $\tilde{v}_1$ and $\tilde{v}_1 \pm \Delta\tilde{v}_1$, respectively, $\Delta\tilde{v}_1 \cong \frac{SD_1^{(j)}}{\overline{ISI}_1^{(j)} - m_1} (\log 10)^{-1} \cong a_1 (\log 10)^{-1}$. Because $a_1 (\log 10)^{-1}$ is a constant and thus does not depend on the event index $j$, $v_1$ has a similar variability for all events (see bursts shown in different colors in Fig 4D). Because $v_1 = \log_{10}(ISI_1[ms] - m_1[ms])$ cannot be defined when $ISI_1 - m_1 \leq 0$, bursts with $ISI_1 \leq m_1$ were removed from the analyses (see below for the potential effect of the exclusion). $v_2$ was similarly defined for $ISI_2$ (Fig 4C, bottom). $v_1$ and $v_2$ were linearly transformed as $v_d^* = c_{1d}v_1 + c_{2d}$ $d = 1, 2$, so that the probability distribution of $v_1^*$ and $v_2^*$ fits to the normal distribution with a mean = 0 and SD = 1, where $c_{1d}$ and $c_{2d}$ are constants. A principal component analysis was then conducted on the distribution of $v_1^*$ and $v_2^*$ (Fig 4D), and the new variables $u_1$ and $u_2$ were defined as follows by scaling $v_1^*$ and $v_2^*$ along these axes, so that the standard deviations along these axes were equal to 1 (Fig 4E). Let $(p_1, p_2)$ and $(q_1, q_2)$ be the unit vectors that are parallel to the principal axes of the distribution of $(v_1^*, v_2^*)$, and $c^2$ and $d^2$ ($c > 0$, $d > 0$) be the corresponding variances. $u_1$ and $u_2$ were defined as follows:

$$\begin{pmatrix} u_1 \\ u_2 \end{pmatrix} = \begin{pmatrix} p_1 & q_1 \\ p_2 & q_2 \end{pmatrix} \begin{pmatrix} c^{-1} & 0 \\ 0 & d^{-1} \end{pmatrix} \begin{pmatrix} p_1 & p_2 \\ q_1 & q_2 \end{pmatrix} \begin{pmatrix} v_1^* - \overline{v_1^*} \\ v_2^* - \overline{v_2^*} \end{pmatrix},$$

where $\overline{v_1^*}$ and $\overline{v_2^*}$ are the averages of $v_1^*$ and $v_2^*$, respectively. The burst phase (Fig 4E) was $\text{Arg}(u_1 + u_2\sqrt{-1})$, where Arg denotes the argument of complex numbers.

**Deviation from 3-BTA.** The deviation from 3-BTA was calculated as follows. Let $N_3$ be the number of 3-spike bursts recorded from the $k$-th cell, and $t_i$ and $\varphi_i$ ($i = 1, \cdots, N_3$) be the time of the first spike and burst phase of the $i$-th 3-spike burst. Three-spike bursts were divided into 12 groups according to the burst phase; the lower and upper limit of the phase of bursts in each group was $0°\pm15°$, $30°\pm15°$, $\cdots$, $330°\pm15°$. Let $B_M$ be the set of index $i$ for which the $i$-th 3-spike burst is in the $M$-th group ($0 \leq M \leq 11$):

$$B_{M,k} = \{i | M \times 30° - 15° \leq \varphi_i < M \times 30° + 15°\}.$$

The BTA for bursts in the $M$-th group, $BTA_{M,k}(\tau)$, was determined via the following calculation:

$$BTA_{M,k}(\tau) = \mathcal{F}_{16} * \left\{ \frac{1}{\#B_{M,k}} \sum_{i \in B_{Mk}} St(t_i + \tau) \right\},$$

where $\mathcal{F}_{16}$ is the smoothing filter defined by Eq 3, $\#B_{M,k}$ is the number of indices in $B_{M,k}$ and $St(t)$ is the stimulus value at time $t$. The $BTA_{M,k}(\tau)$ ($1 \leq M \leq 12$) of the cell used in Fig 4A is shown in Fig 5A. The deviation from 3-BTA is calculated as follows:

$$Dev_{M,k}(\tau) = BTA_{M,k}(\tau) - 3\text{-}BTA_k(\tau),$$

where $3\text{-}BTA_k(\tau)$ is the 3-BTA for the $k$-th cell. The $Dev_{M,k}(\tau)$ of the cell used in Fig 4A is shown in Fig 5C. Let $N_{cells}$ be the number of cells. The average and SEM of $Dev_{M,k}(\tau)$ was

calculated as follows:

$$\text{Dev}_{M,ave}(\tau) = \frac{1}{N_{\text{cells}}} \sum_{k=1}^{N_{\text{cells}}} \text{Dev}_{M,k}(\tau)$$

$$\text{Dev}_{M,\text{SEM}}(\tau) = \frac{1}{\sqrt{N_{\text{cells}}}} \left[ \frac{1}{N_{\text{cells}}} \sum_{k=1}^{N_{\text{cells}}} \left\{ \text{Dev}_{M,k}(\tau) - \text{Dev}_{M,ave}(\tau) \right\}^2 \right]^{\frac{1}{2}}.$$

Fig 5D shows $\text{Dev}_{M,ave}(\tau)$, and Fig 5E shows $\text{Dev}_{M,ave}(\tau)$ and $\text{Dev}_{M,\text{SEM}}(\tau)$ for $M = 0, 3, 6, 9$.

**Test of the effect of removing 3-spike bursts with short ISIs.** Whether the exclusion of bursts with small ISIs biases the analyses was tested using a decoding scheme that included bursts with small ISIs. For each cell, we collected all bursts that were not excluded. Among these bursts, $\text{ISI}_{10}$ and $\text{ISI}_{20}$, the smallest $\text{ISI}_1$ ($>m_1$) and smallest $\text{ISI}_2$ ($>m_2$), were determined, where $m_1$ and $m_1$ were the intersects determined as described for Fig 4C. The $\text{ISI}_1$ and $\text{ISI}_2$ values of all the excluded bursts were replaced with $\text{ISI}_{10}$ and $\text{ISI}_{20}$, respectively, and these bursts were incorporated into the analysis by transforming their coordinates as described above. The results (S3M–S3O Fig) were almost indistinguishable from the original findings (Fig 5A, 5D and 5E).

**Linear reconstruction.** Let $t_1^{(l)}, t_2^{(l)}$, and $t_3^{(l)}$ be the time of the first, second, and third spikes in the $l$-th 3-spike burst, where $l = 1, \cdots, N_{3\text{-spike}}$ and $N_{3\text{-spike}}$ is the number of all 3-spike bursts recorded from a cell. Let $St(t)$ be the light intensity at time $t$ (mean = 0, SD = 1). $\text{STA}_{3\text{-spike}}(\tau)$ ($-300$ ms$\leq\tau\leq 0$ ms) was determined by calculating

$$\text{STA}_{3\text{-spike}}(\tau) = \mathcal{F}_{16} * \left\{ \frac{1}{3} \frac{1}{3N_{3\text{-spike}}} \sum_{l=1}^{N_{3\text{-spike}}} \sum_{g=1}^{3} St(t_g^{(l)} + \tau) \right\},$$

where $\mathcal{F}_{16}$ is the smoothing filter defined by Eq 3. The factor $\frac{1}{3}$ was applied so that the simplest reconstruction, $3\times\text{STA}_{3\text{-spike}}(\tau)$, had an amplitude that was similar to that of 3-BTA. For $\tau<-300$ ms and $0$ ms $<\tau$, $\text{STA}_{3\text{-spike}}(\tau) = 0$. The linear reconstruction for the $l$-th 3-spike burst was:

$$\text{Recon}^{(l)}(\tau) = \text{STA}_{3\text{-spike}}(\tau) + \text{STA}_{3\text{-spike}}(\tau - (t_2^{(l)} - t_1^{(l)})) + \text{STA}_{3\text{-spike}}(\tau - (t_3^{(l)} - t_1^{(l)})).$$

$\text{Recon}^{(l)}(\tau)$ for a simulated burst is shown in Fig 7A. $\text{Recon}^{(l)}(\tau)$ was calculated for all 3-spike bursts ($l = 1, \cdots, N_{3\text{-spike}}$) and subjected to the same analysis that was applied for the actual preceding sequences (compare Fig 5A with 7B, and 5E with 7C).

**Relationship between the 2-spike burst ISI and 3-spike burst $u_1$ and $u_2$ values.** For each cell, the relationship between the 2-spike burst ISI and 3-spike burst $u_1$ and $u_2$ values was analyzed as follows. Let $N_{\text{Event }2,3}$ be the number of events in which at least one 2-spike burst and one 3-spike burst occurred. For each event $j$ ($j = 1, \cdots, N_{\text{Event }2,3}$), the average intraburst ISIs of the 2-spike bursts ($m^{(j)}$), and the average of $u_1$ and $u_2$ ($\overline{u_1}^{(j)}$ and $\overline{u_2}^{(j)}$) were determined. The values $(m^{(j)}, \overline{u_1}^{(j)}, \overline{u_2}^{(j)})$ ($j = 1, \cdots, N_{\text{Event2,3}}$) were fitted as $m^{(j)} = \alpha_0 + \alpha_1 \overline{u_1}^{(j)} + \alpha_2 \overline{u_2}^{(j)}$, where $\alpha_0, \alpha_1$, and $\alpha_2$ were constants. The direction of the steepest gradient (Fig 8B) was that of the vector $(\alpha_1, \alpha_2)$, i.e., $\text{Arg}(\alpha_1 + \alpha_2\sqrt{-1})$, where Arg denotes the argument of complex numbers.

**Approximately independent components.** The approximately independent components $w_1$ and $w_2$ were defined as follows. Let $N_3$ be the number of 3-spike bursts recorded from the $k$-th cell, and $u_1^{(i)}$ and $u_2^{(i)}$ be the $u_1$ and $u_2$ values of the $i$-th 3-spike burst ($i = 1, 2, \cdots, N_3$), respectively. Let $j(i)$ be the event index of the $i$-th 3-spike burst ($j(i) = 1, 2, \cdots, N_{\text{Event}}$, where $N_{\text{Event}}$ is the number of events). Let $\overline{u_1}^{(j)}$ and $\overline{u_2}^{(j)}$ be the average of $u_1^{(i)}$ and $u_2^{(i)}$ in the $j$-th event.

The trial-to-trial variation was defined as:

$$\left(u_1^{(i)} - \overline{u_1}^{(j(i))}, u_2^{(i)} - \overline{u_2}^{(j(i))}\right),$$

where $i = 1, 2, \cdots, N_3$; and its distribution is shown in Fig 8C. The principal component analysis was conducted on this distribution (yellow and green lines in Fig 8C). Let $\theta$ ($0° \leq \theta < 180°$) be the angle between the $u_1$ axis and the principal axis with the smaller variance (Fig 8C and 8E). The two approximately independent components, $w_1$ and $w_2$, were as follows:

$$\begin{pmatrix} w_1 \\ w_2 \end{pmatrix} = \begin{pmatrix} \cos\theta & \sin\theta \\ \cos(\theta + 90°) & \sin(\theta + 90°) \end{pmatrix} \begin{pmatrix} u_1 \\ u_2 \end{pmatrix}$$

Let $\theta_k$ be the $\theta$ value of the $k$-th cell. Let $N_3$ be the number of 3-spike bursts recorded from the cell, and $t_i$ and $\varphi_i$ ($i = 1, \cdots, N_3$) be the time of the first spike and the burst phase of the $i$-th 3-spike burst, respectively. Let

$$C_{1+,k} = \{i | \theta_k - 15° \leq \varphi_i < \theta_k + 15°\},$$

$$C_{2+,k} = \{i | (\theta_k + 90°) - 15° \leq \varphi_i < (\theta_k + 90°) + 15°\},$$

$$C_{1-,k} = \{i | (\theta_k + 180°) - 15° \leq \varphi_i < (\theta_k + 180°) + 15°\}, \text{ and}$$

$$C_{2-,k} = \{i | (\theta_k + 270°) - 15° \leq \varphi_i < (\theta_k + 270°) + 15°\}.$$

The deviation from 3-BTA for bursts with the phase $\theta_k$, $\theta_k+90°$, $\theta_k+180°$, $\theta_k+270°$, were defined using the following equations ($d = 1, 2$):

$$\text{Dev}_{d+,k}(\tau) = \mathcal{F}_{16} * \left\{ \frac{1}{\#C_{d+,k}} \sum_{i \in C_{d+,k}} St(t_i + \tau) \right\} - \text{3-BTA}_k(\tau) \text{ and}$$

$$\text{Dev}_{d-,k}(\tau) = \mathcal{F}_{16} * \left\{ \frac{1}{\#C_{d-,k}} \sum_{i \in C_{d-,k}} St(t_i + \tau) \right\} - \text{3-BTA}_k(\tau),$$

where $d = 1, 2$, $\mathcal{F}_{16}$ is the smoothing filter defined by Eq 3, and $\#C_{d+,k}$ and, $\#C_{d-,k}$ represent the number of indices in $C_{d+,k}$ and $C_{d-,k}$, respectively. $St(t)$ is the light intensity at time $t$, and 3-BTA$_k(\tau)$ is the 3-BTA for the $k$-th cell. $\text{Dev}_{1+,k}(\tau) - \text{Dev}_{1-,k}(\tau)$ and $\text{Dev}_{2+,k}(\tau) - \text{Dev}_{2-,k}(\tau)$ are shown in S2B and S2C Fig, respectively. The average and SEM across cells were calculated as follows:

$$\text{Dev}_{d+,\text{ave}}(\tau) = \frac{1}{N_{\text{cells}}} \sum_{k=1}^{N_{\text{cells}}} \text{Dev}_{d+,k}(\tau)$$

$$\text{Dev}_{d+,\text{SEM}}(\tau) = \frac{1}{\sqrt{N_{\text{cells}}}} \left[ \frac{1}{N_{\text{cells}}} \sum_{k=1}^{N_{\text{cells}}} \{\text{Dev}_{d+,k}(\tau) - \text{Dev}_{d+,\text{ave}}(\tau)\}^2 \right]^{\frac{1}{2}},$$

where $N_{\text{cells}}$ is the number of cells. $\text{Dev}_{1+,\text{ave}}(\tau)$, $\text{Dev}_{1+,\text{SEM}}(\tau)$, $\text{Dev}_{2+,\text{ave}}(\tau)$, and $\text{Dev}_{2+,\text{SEM}}(\tau)$ are shown in Fig 8F.

To calculate the projection,

$$D_{1+,k} = \{i | \theta_k - 45° \leq \varphi_i < \theta_k + 45°\}$$

$$D_{2+,k} = \{i | (\theta_k + 90°) - 45° \leq \varphi_i < (\theta_k + 90°) + 45°\}$$

$$D_{1-,k} = \{i | (\theta_k + 180°) - 45° \leq \varphi_i < (\theta_k + 180°) + 45°\}$$

$$D_{2-,k} = \{i | (\theta_k + 270°) - 45° \leq \varphi_i < (\theta_k + 270°) + 45°\}$$

were defined. Note that this definition divides all bursts into four groups. For $d = 1, 2$,

$$\text{Dev}_{90,d+,k}(\tau) = \mathcal{F}_{16} * \left\{ \frac{1}{\#D_{d+,k}} \sum_{i \in D_{d+,k}} St(t_i + \tau) \right\} - 3\text{-BTA}_k(\tau),$$

$$\text{Dev}_{90,d-,k}(\tau) = \mathcal{F}_{16} * \left\{ \frac{1}{\#D_{d-,k}} \sum_{i \in D_{d-,k}} St(t_i + \tau) \right\} - 3\text{-BTA}_k(\tau), \text{ and}$$

$$f_d(t) = \text{Dev}_{90,d+,k}(\tau) - \text{Dev}_{90,d-,k}(\tau)$$

were defined, and the raw projection value [27],

$$s^*_{w1,k}(t) = \frac{1}{300 \text{ ms}} \int_{-300 \text{ ms}}^{0 \text{ ms}} St(t + \tau) f_1(\tau) d\tau,$$

was collected at all 3-spike bursts and normalized to define and process $s_{w1,3\text{-spike},k}$ similarly to $s_{D2,\text{burst},k}$. Finally, $s_{w2,3\text{-spike}}$ was similarly defined using $f_2(t)$.

## Supporting information

**S1 Fig. Identification of bursts and events. (A–C)** Identification of bursts. **(A)** ISI histogram. $T_{\text{thresh}}$ indicates the threshold interval, which was set at the trough between the two peaks in the histogram. **(B)** Algorithm used to define bursts. The vertical lines represent spikes. When two consecutive spikes occurred with an interval shorter than $T_{\text{thresh}}$, they were incorporated into the same burst. If the interval was longer than $T_{\text{thresh}}$, the two consecutive spikes were separated into two different bursts. **(C)** Rates of isolated spikes (1) and bursts with 2–7 spikes (2–7) plotted against the threshold interval. Data from the cell shown in Fig 1. **(D–F)** Identification of events. **(D)** Algorithm used to define events. The top panel shows a schematic raster plot. Each row shows spikes that occurred during a single repeat of the stimulation. The first spikes of the bursts (black lines at the top) were merged into a single train (middle). When the two first spikes in the merged train were closer than $T_{\text{thresh}}$, they were incorporated into the same event; otherwise, they were assigned into two different events (bottom). **(E)** The intervals of the merged train of first spikes were determined and their histogram is shown. $T_{\text{thresh}}$ indicates the threshold interval. Data from the cell shown in Fig 1. **(F)** Exceptional case of the event identification. The top panel presents a schematic raster plot. Each row shows spikes that occurred during a single repeat of the stimulus. The black lines in the top panel represent the first spikes of bursts. The timing of the bursts shows a large jitter in different repeats. The asterisks indicate two consecutive bursts. The middle panel shows the merged train of the first spikes. Because all intervals were $<T_{\text{thresh}}$, all bursts were incorporated into the same event

(bottom). The two consecutive bursts marked by the asterisks in the top panel were merged into one burst. See also Materials and Methods.
(TIF)

**S2 Fig. Stimulus features encoded by the burst patterns of individual neurons. (A)** Stimulus sequences encoded by 2-spike burst ISIs. Each row represents one of the 19 cells that generated at least 1500 2-spike bursts. For each cell, the deviation from 2-BTA was calculated by subtracting 2-BTA from the average of the stimulus sequence preceding the 2-spike bursts with the longest and shortest 50% of intraburst ISIs (see yellow and blue lines in Fig 3G, respectively). The difference in the deviation from 2-BTA for the longest and shortest ISIs are shown. **(B and C)** Stimulus sequences encoded by the two approximately independent components of 3-spike bursts. Data for the 19 cells that generated at least 1200 3-spike bursts. (B). The $k$-th row represents $Dev_{1+,k}(\tau)-Dev_{1-,k}(\tau)$, *i.e.*, the difference in the average stimulus sequence preceding 3-spike bursts with the burst phase within the range centered by $\theta$ and $\theta + 180°$, where $\theta$ represents the angle between the $u_1$ axis and the principle axis with the smaller variance (see Fig 8E). (C) The $k$-th row represents $Dev_{2+,k}(\tau)-Dev_{2-,k}(\tau)$, *i.e.*, the difference in the average stimulus sequence preceding 3-spike bursts with the burst phase within the range centered by $\theta + 90°$ and $\theta + 270°$, calculated for the $k$-th cell. See Materials and Methods.
(TIF)

**S3 Fig. Effect of the details of the 3-spike burst analysis. (A–L)** Analysis of the effect of the stimulus alignment. **(A–C)** Stimulus sequences were aligned on the second spike in bursts. **(A)** Bursts were grouped according to the burst phase with the binwidth of 30°. Stimulus sequences preceding bursts in each group were averaged. Data from the cell shown in Fig 5A (compare with Fig 5A). **(B)** 3-BTA was calculated using preceding stimulus sequences aligned on the second spike in bursts. The deviation of 3-BTA was calculated by subtracting the 3-BTA from the data in (A). The panel shows the deviation of 3-BTA averaged across the 19 cells that generated at least 1200 3-spike bursts. Compare with Fig 5D. **(C)** The thick lines indicate the deviation of 3-BTA at the indicated burst phases averaged across the 19 cells. The thin lines are the SEM calculated across the cells. Compare with Fig 5E. **(D–F)** Analysis similar to that described in (A–C), with the exception that the stimulus sequences were aligned on the third spike in bursts. **(G–I)** Analysis similar to that described in (A–C), with the exception that the stimulus sequences were aligned on the middle of the duration of bursts, i.e., the middle between the first and third spikes. **(J–L)** Stimulus sequences encoded by the independent components $w_1$ and $w_2$. Stimulus sequences were aligned on the second spike (J), the third spike (K), and the middle of the duration of the bursts (L). For each cell, the deviations from 3-BTA were calculated by subtracting 3-BTA from the average stimulus sequence preceding 3-spike bursts with the burst phase within the range centered by $\theta$ (for $w_1$) and $\theta + 90°$ (for $w_2$), where $\theta$ represents the angle between the $u_1$ axis and the principle axis with the smaller variance (see Fig 8E). The thick lines indicate the deviation from 3-BTA averaged across 19 cells that generated at least 1200 3-spike bursts (yellow: $w_1$; green: $w_2$). The thin lines show SEM values calculated across the cells. **(M–O)** Bursts excluded from the analysis because of small ISIs were incorporated into the analysis by replacing the ISIs with the smallest ISIs of the incorporated bursts. Analysis similar to that described in (A–C), with the exception that the stimulus sequences were aligned on the first spike of bursts. Compare with Fig 5E.
(TIF)

**S4 Fig. Effect of stimulus alignment on the analysis using the natural scene movie.** The deviation from 3-BTA was calculated by aligning bursts on the second spike (A), third spike (B), and middle of the duration (C) of bursts. For each cell, the deviation from 3-BTA was

calculated by subtracting 3-BTA from the average stimulus sequence preceding 3-spike bursts with the burst phase within a range centered by 0˚ (red), 90˚ (green), 180˚ (cyan), and 270˚ (magenta). The average (thick lines) and SEM (thin lines) among the 8 cells that generated more than 1000 3-spike bursts are shown. Compare with Fig 6D.
(TIF)

## Acknowledgments

We thank Markus Meister, Takao K. Hensch, Shun-ichi Amari, Shin Yanagihara, Neal Hessler, Yoshihiro Yoshihara, Stephan A. Baccus, Bence P. Ölveczky, Nick Lesica, Xin Jing, Charles Yokoyama, Naoki Masuda, Shiro Ikeda, Makoto Kaneda, Tomoki Fukai, Keita Watanabe, and Tetsuya Haga for discussion.

## Author Contributions

**Conceptualization:** Toshihiko Hosoya.

**Data curation:** Toshiyuki Ishii.

**Formal analysis:** Toshiyuki Ishii, Toshihiko Hosoya.

**Funding acquisition:** Toshihiko Hosoya.

**Investigation:** Toshihiko Hosoya.

**Methodology:** Toshihiko Hosoya.

**Project administration:** Toshihiko Hosoya.

**Software:** Toshihiko Hosoya.

**Supervision:** Toshihiko Hosoya.

**Validation:** Toshihiko Hosoya.

**Visualization:** Toshiyuki Ishii, Toshihiko Hosoya.

**Writing – original draft:** Toshiyuki Ishii, Toshihiko Hosoya.

**Writing – review & editing:** Toshihiko Hosoya.

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
