## [Decision Letter · Decision Letter 0]

23 Mar 2020

Dear Dr. Hosoya,

Thank you very much for submitting your manuscript "Interspike intervals within retinal spike bursts combinatorially encode multiple stimulus features" for consideration at PLOS Computational Biology.

As with all papers reviewed by the journal, your manuscript was reviewed by members of the editorial board and by several independent reviewers. In light of the reviews (below this email), we would like to invite the resubmission of a significantly-revised version that takes into account the reviewers' comments.

The paper presents potentially interesting results. However, two of the reviewers raised major concerns, which need to be addressed through appropriate additional research work and text revisions.

We cannot make any decision about publication until we have seen the revised manuscript and your response to the reviewers' comments. Your revised manuscript is also likely to be sent to reviewers for further evaluation.

Sincerely,

Stefano Panzeri

Guest Editor

PLOS Computational Biology

Daniele Marinazzo

Deputy Editor

PLOS Computational Biology

The paper presents potentially interesting results. However, two of the reviewers raised major concerns, which need to be addressed through appropriate additional research work and text revisions.

Reviewer's Responses to Questions

**Comments to the Authors:**

Reviewer #1: In their manuscript "Interspike intervals within retinal spike bursts combinatorially encode multiple stimulus features" the authors analyze the relation between intraburst ISIs of 2-spike or 3-spike bursts and stimulus features on data recorded from salamander retinal ganglion cells. The authors identify stimulus features encoded by the different ISIs.

The analysis is timely and very naturally extends previous studies which showed that the number of spikes within a burst, the burst onset and duration of a burst carry stimulus information. The manuscript is very well written and the figures are clear and informative.

Major point:

There is a conceptual issue with the mutual information analysis. A couple of times (lines 98, 116, 152), the authors mention the "information about the stimulus". Intuitively, I would expect that stimulus here is referring to the light intensity profile. But from Materials and Methods, it becomes clear that the authors are referring to information about "event" identity. The authors define "events" as reproducible bursts. Essentially, an event is not an external stimulus variable but itself a neural response property which happens to be linked to the external stimulus onset. However, even for constant stimuli, RGCs can elicit complex patterns for hundreds of milliseconds similar to the patterns shown in Fig 1. Thus, there seems to be a conflation of the firing dynamics of the RGCs and the light intensity profile in the mutual information analysis. Therefore, it seems the authors show that they can distinguish between different time points relative to a stimulus onset but not necessarily between different stimuli i.e. between different intensity profiles or between different intensities within a profile. In my opinion, this is not "information about the stimulus". This concerns the mutual information analysis only. The authors also evaluate relations of intraburst ISIs to amplitudes and temporal phases of oscillatory components which I find convincing.

Minor points:

The authors analyzed 2-spike and 3-spike bursts only and state that bursts with 4 or more spikes were too rare to conduct a similar analysis (line 314 and Fig 1D, though the illustrated green reproducible burst in Fig 1A then appears to be unusual). I was wondering, could anything be said when pooling together all bursts with more than 3 spikes? Do the first two intraburst ISIs hint at the same characteristics as for the 3-spike bursts? An answer to this question would shed light on the general validity of the amplitude and phase encoding.

Line 480: typo: insert "in" before "the open"

Reviewer #2: This is a very interesting study supporting the multiplexed encoding of stimulus information in a combinatorial burst code. The authors analysed data from retinal ganglion cells of salamander larvae when stimulated with oscillatory light stimuli and natural scenes. The main result is that intra-burst ISIs encode faster features of the stimuli in comparison with spike number alone. Moreover, the study shows that ISIs can combinatorially encode different features of the stimulus. The analyses are done with great attention to detail and there is an overall excellent quality of presentation.

Reviewer #3: review is uploaded as an attachment.

**Have all data underlying the figures and results presented in the manuscript been provided?**

Reviewer #1: No: The authors stated that they will provide data and code on acceptance.

Reviewer #2: Yes

Reviewer #3: Yes

PLOS authors have the option to publish the peer review history of their article (what does this mean?). If published, this will include your full peer review and any attached files.

Reviewer #1: No

Reviewer #2: No

Reviewer #3: Yes: Fleur Zeldenrust
---

## [Decision Letter · Decision Letter 1]

17 Aug 2020

Dear Dr. Hosoya,

Thank you very much for submitting your manuscript "Interspike intervals within retinal spike bursts combinatorially encode multiple stimulus features" for consideration at PLOS Computational Biology. As with all papers reviewed by the journal, your manuscript was reviewed by members of the editorial board and by several independent reviewers. The reviewers appreciated the attention to an important topic. Based on the reviews, we are likely to accept this manuscript for publication, providing that you modify the manuscript (only minor text edits) according to the review recommendations.

Sincerely,

Stefano Panzeri

Guest Editor

PLOS Computational Biology

Daniele Marinazzo

Deputy Editor

PLOS Computational Biology

[LINK]

Reviewer's Responses to Questions

**Comments to the Authors:**

Reviewer #1: The authors revised their manuscript extensively and satisfactorily addressed all of my concerns. In particular, they clarified the relation between ISI burst patterns and stimuli as opposed to event identity by revising their estimator. Their results are still qualitatively similar, supporting the robustness of their findings. They also performed a more extensive analysis of the contributions of different ISIs, yielding interesting results.

Reviewer #3: See pdf

**Have all data underlying the figures and results presented in the manuscript been provided?**

Reviewer #1: Yes

Reviewer #3: Yes

PLOS authors have the option to publish the peer review history of their article (what does this mean?). If published, this will include your full peer review and any attached files.

Reviewer #1: No

Reviewer #3: **Yes: **Fleur Zeldenrust
---

## [Editor Report · Decision Letter 2]

22 Sep 2020

Dear Dr. Hosoya,

We are pleased to inform you that your manuscript 'Interspike intervals within retinal spike bursts combinatorially encode multiple stimulus features' has been provisionally accepted for publication in PLOS Computational Biology.

Best regards,

Stefano Panzeri

Guest Editor

PLOS Computational Biology

Daniele Marinazzo

Deputy Editor

PLOS Computational Biology

---

## [Editor Report · Acceptance letter]

26 Oct 2020

PCOMPBIOL-D-20-00205R2 

Interspike intervals within retinal spike bursts combinatorially encode multiple stimulus features

Dear Dr Hosoya,

I am pleased to inform you that your manuscript has been formally accepted for publication in PLOS Computational Biology. Your manuscript is now with our production department and you will be notified of the publication date in due course.

With kind regards,

Matt Lyles
